# Quantitative comparison of geological data and model simulations constrains early Cambrian geography and climate

Thomas W. Wong Hearing [1,2 ✉], Alexandre Pohl[3,4 ✉], Mark Williams [2 ✉], Yannick Donnadieu [5], Thomas H. P. Harvey[2], Christopher R. Scotese[6], Pierre Sepulchre [7], Alain Franc [8,9] & Thijs R. A. Vandenbroucke [1]

Marine ecosystems with a diverse range of animal groups became established during the early Cambrian (~541 to ~509 Ma). However, Earth's environmental parameters and palaeogeography in this interval of major macro-evolutionary change remain poorly constrained. Here, we test contrasting hypotheses of continental configuration and climate that have profound implications for interpreting Cambrian environmental proxies. We integrate general circulation models and geological observations to test three variants of the 'Antarctocentric' paradigm, with a southern polar continent, and an 'equatorial' configuration that lacks polar continents. This quantitative framework can be applied to other deep-time intervals when environmental proxy data are scarce. Our results show that the Antarctocentric palaeogeographic paradigm can reconcile geological data and simulated Cambrian climate. Our analyses indicate a greenhouse climate during the Cambrian animal radiation, with mean annual sea-surface temperatures between ~9 °C to ~19 °C and ~30 °C to ~38 °C for polar and tropical palaeolatitudes, respectively.

[1] Department of Geology, Ghent University, Ghent, Belgium. [2] School of Geography, Geology and the Environment, University of Leicester, Leicester, UK. [3] Department of Earth and Planetary Sciences, University of California, Riverside, CA, USA. [4] Biogéosciences, UMR 6282, UBFC/CNRS, Université Bourgogne Franche-Comté, Dijon, France. [5] Aix-Marseille Univ, CNRS, IRD, INRA, Coll. France, CEREGE, Aix-en-Provence, France. [6] Department of Earth & Planetary Sciences, Northwestern University, Evanston, IL, USA. [7] Laboratoire des Sciences du Climat et de l'Environnement, LSCE/IPSL, CEA-CNRS-UVSQ, Université Paris-Saclay, Gif-sur-Yvette, France. [8] INRAE, University of Bordeaux, BIOGECO, Cestas, France. [9] Inria Bordeaux-Sud-Ouest, Pleiade, Talence, France. ✉email: thomas.wonghearing@ugent.be; alexandre.pohl@u-bourgogne.fr; mri@leicester.ac.uk

Animal-rich ecosystems evolved in Earth's oceans over the Neoproterozoic–Phanerozoic transition approximately half a billion years ago, becoming firmly established in the early Cambrian animal radiation[1,2]. Extensive efforts have been put into examining changes in Earth's geochemical cycles and global oxygen levels over this interval[3–6]. However, much about the earliest Phanerozoic world remains uncertain, especially the state of Earth's climate[7,8] and the configuration of Earth's continents (Fig. 1), e.g. ref. [9].

Quantitative palaeoclimate proxy data from stable oxygen isotopes are becoming available for the early Cambrian (Terreneuvian and Epoch 2, ~541 to ~509 Ma)[7,8]. However, these geochemical proxy data only provide spatio-temporal snapshots into Earth's climate, and are thus most powerful when incorporated into a holistic global view of the available evidence. The spatial distributions of fossil plankton have been used to reconstruct early Palaeozoic climatic belts[10], but the limited development of planktic ecosystems in the early Cambrian hinders their use here[11].

Climatically sensitive lithologies, such as evaporitic, lateritic, and glaciogenic deposits, are another valuable source of ancient climate data and can provide greater temporal and palaeogeographic coverage than can geochemical proxy data. Climatically sensitive lithologies have been used to reconstruct Earth's climate throughout the Phanerozoic, e.g. ref. [12], and have been employed with considerable success to evaluate the predictions of climate models[13–17]. In particular, we examine five categories of climatically sensitive lithology capturing a broad spectrum of climatic conditions: evaporites and calcretes form in arid and semi-arid settings, glaciogenic deposits form in cold (freezing) conditions, whilst laterites are characteristic products of continental weathering under hot humid conditions, and oolitic limestones form in warm shallow marine settings where the prevailing continental climate is arid to semi-arid (see ref. [12], 'Methods', and references in Table 1). Coupling climatically sensitive lithologies and climate model simulations provides a robust method to reconstruct deep-time climate conditions on a global scale.

Accurate early Cambrian reconstructions of the configuration of Earth's continents, typically derived from palaeomagnetic and palaeobiogeographic data, are required to compare geological data with climate model simulations. However, Cambrian palaeomagnetic data often come with substantial spatial and temporal uncertainties[18], and palaeobiogeographic methods are hindered by the endemism of many early Cambrian faunas[19,20]. Consequently, there is latitude for debate over the positions of even major continents in the early Cambrian, e.g. ref. [9]. The result of these uncertainties is a suite of disparate hypotheses of early Cambrian continental configuration in the published literature, e.g. refs. [18,19,21–33]. These include variants of the 'Antarctocentric' paradigm[25–28,30,31], characterised by a portion of West Gondwana lying at or near the South Pole, and an 'equatorial' configuration[18,19,21,33] in which all the major palaeocontinents, including West Gondwana, occupy low to mid-palaeolatitudes (Fig. 1). The Antarctocentric configurations are mainly constructed using palaeomagnetic and palaeobiogeographic data[25–28,30,31]. In contrast, the equatorial configurations primarily derive from qualitative expectations of the distribution of climatically sensitive lithologies, notably the occurrence of lower Cambrian limestones and evaporites in North Africa[18,19,33,34]. However, the equatorial model requires rapid continental reconfiguration through the Cambrian Period, including an order-of-magnitude change in the rate of movement of Gondwana, to be reconciled with Ordovician palaeogeographies that are well-constrained by palaeomagnetic, palaeobiogeographic, and sedimentological data, e.g. refs. [27,31].

In this study, we quantitatively test the congruence of general circulation model (GCM) climate simulations with the distribution of climatically sensitive lithologies for four coeval hypotheses of continental configuration. We show that an equatorial continental configuration is not required to reconcile the observed distribution of climatically sensitive lithologies with simulated Cambrian climate, and consequently that unusually rapid plate tectonic movements are not required to explain the distribution of early Cambrian climatically sensitive lithologies. Furthermore, we provide testable constraints on early Cambrian climate and geography, and suggest that our approach can be used to constrain first-order assessments of key boundary conditions, including atmospheric greenhouse gas forcing, in deep time.

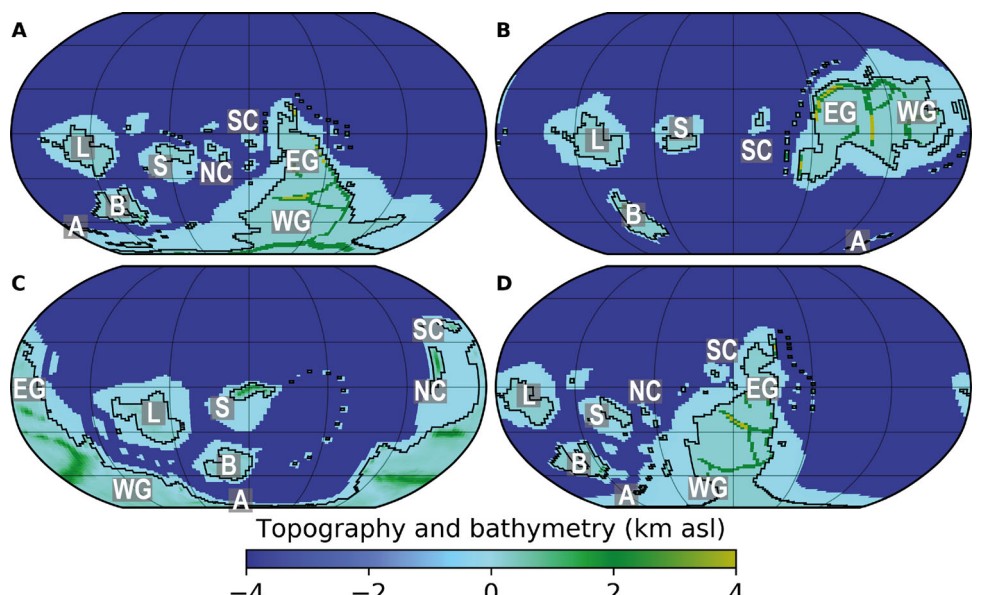

**Fig. 1 Four continental configurations (A–D) tested here, overlain with the palaeotopography and palaeobathymetry used in our FOAM simulations.** Configuration A: 510 Ma reconstruction, after refs. [25–27]. Configuration B: terminal Ediacaran–early Cambrian reconstruction, after refs. [18,19]. Configuration C: 510 Ma reconstruction, after refs. [30,31]. Configuration D: Cambrian reconstruction, after ref. [28]. A: Avalonia; B: Baltica; EG: East Gondwana; L: Laurentia; NC: North China; S: Siberia; SC: South China; WG: West Gondwana; asl: above sea level.

**Table 1 Formation conditions of climatically sensitive lithologies and the categories used in this study.**

| Lithology | Category[a] | Weight[b] | Formation conditions[c] | References |
|---|---|---|---|---|
| Calcretes | C | 1.00 | Persistent semi-arid to arid continental conditions on millennial timescales, with fluctuating groundwater level. Evaporation > precipitation at least 6 months in every year; 140 mm < MAP <1400 mm. | Refs. 73,74 |
| Evaporites (general) | E | 1.00 | Persistent arid to semi-arid conditions, where evaporation typically exceeds precipitation. Quaternary evaporite deposits are observed to form under fully arid and seasonally arid conditions, under polar to tropical temperatures. | Refs. 12,69,74–76 |
| Anhydrite | E | 1.00 | Arid to semi-arid conditions, often within a (partially) isolated basin. As for general evaporites. | Refs. 12,69 |
| Gypsum | E | 1.00 | Arid to semi-arid conditions, often within a (partially) isolated basin. As for general evaporites. Most though not all modern gypsum forms under conditions of MAP < 400 mm and evaporation > precipitation in every month of the year. | Refs. 12,69,74,75 |
| Halite | E | 1.00 | Arid to semi-arid conditions, often within a (partially) isolated basin. As for general evaporites. | Refs. 12,69 |
| Evaporite pseudomorphs | E | 0.75 | Arid to semi-arid conditions, often within a (partially) isolated basin. As for general evaporites. Down-weighted due to reliance on interpreting the original morphology of the pseudomorphs. | Refs. 12,69,74,75 |
| Length-slow chalcedony | E | 1.00 | Arid to semi-arid alkaline soils; forms in association with sulfate and halite products. | Refs. 69,77 |
| Striated dropstones | G | 1.00 | Cold (freezing) conditions nearby. Outsized clasts may be down-weighted if glacial origin is uncertain. | Refs. 12,78 |
| Tillite | G | 1.00 | Cold (freezing) conditions with significant land ice present. Diamictite texture may be down-weighted if glacial origin is uncertain. | Refs. 12,78 |
| Bauxite | L | 1.00 | Intense continental weathering under high humidity and temperature; evaporation < precipitation in 9 to 11 months of the year; MAAT > 25 °C and MAP >1800 mm, or MAAT > 22 °C and MAP >1200 mm. | Refs. 74,79,80 |
| Kaolinite | L | 1.00 | Intense continental weathering under high humidity and typically high temperature, though temperate-region kaolinite deposits are known; low pH is also required and is commonly derived from the presence of humic acids. | Refs. 12,81 |
| Laterite | L | 1.00 | Intense continental weathering under high humidity and temperature; evaporation < precipitation in 9–11 months of the year; MAAT > 25 °C. | Refs. 12,74,82 |
| Oolitic ironstones (berthierine, chamosite) | L | 1.00 | Proximal to intense continental weathering under high humidity and temperature; energetic shallow marine setting. Oolitic ironstones may be down-weighted if the primary mineralogy is uncertain. | Refs. 12,83 |
| Oolitic limestone | O | 1.00 | Warm, (sub)tropical energetic shallow marine setting; starved of clastic sediment input; (super)saturated with respect to calcium carbonate, which supersaturation may be aided by an arid climate. | Refs. 84–86 |

[a]Categories used in our analyses. C = calcrete; E = evaporite; G = glaciogenic; L = lateritic products; O = oolitic limestone.
[b]The weighting listed here is the general case for each lithology. Any individual occurrence may be modified up or down due to its precise conditions.
[c]MAP mean annual precipitation; MAAT mean annual surface air temperature.

## Results

Our lithology database builds on the compilation of ref. [12] with refined geographic and stratigraphic constraints and additional data from the published literature (see 'Methods' and Supplementary Data 1). The four continental configurations tested here include three Antarctocentric configurations in which either the North African (configuration A[25–27]) or South American (configurations C[30,31] and D[28]) part of West Gondwana lies at the South Pole (Fig. 1). The fourth reconstruction is a radically different equatorial model (configuration B[18,19]) in which all of West Gondwana occupies low palaeolatitudes (Fig. 1). We use the Fast Ocean Atmosphere Model (FOAM)[35] GCM to simulate climatic conditions for each of four continental configurations under varying boundary conditions. Because there are currently no reliable and precise $p\mathrm{CO}_2$ proxy data for the Cambrian Period[36–38], we examine a range of atmospheric $p\mathrm{CO}_2$ levels that span the range of estimates from long-term carbon cycle models[37,39–41]. In particular, we run GCM simulations with $p\mathrm{CO}_2$ values of 4, 8, 16, 32, 64, and 128 times the preindustrial atmospheric level (PAL = 280 ppm; see 'Methods' and Supplementary Information). The FOAM GCM is frequently applied in deep-time palaeoclimate studies[42–45], and is particularly well-suited to our purpose because its high throughput means that it can be used to produce the large number of simulations needed to test the wide range of boundary conditions plausible for the early Cambrian. However, as with most earlier generation GCMs, FOAM has a relatively low equilibrium climate sensitivity and consequently tends to simulate colder average temperatures for the same $p\mathrm{CO}_2$ forcing than most recent models used in the Coupled Model Intercomparison Project (CMIP)[46]. Therefore, we hope that future work will be possible using CMIP-class models in the early Palaeozoic context, when Earth System boundary conditions for this interval are better constrained.

To directly compare our GCM simulations and lithology data, we apply the Köppen–Geiger climate classification scheme. The Köppen–Geiger scheme uses temperature and precipitation–evaporation balance thresholds to define discrete climate classes[47,48] (see 'Methods' and Supplementary Table 5), meaning that it can be directly applied to FOAM simulation output data. From data in the published literature, we identified the Köppen–Geiger climate classes under which climatically sensitive lithologies may form. To compare the lithology and GCM datasets, we apply the data–model agreement scoring equation of ref. [49], modified to account for small errors in palaeogeography, to determine which continental configurations and GCM boundary conditions best match the observed distribution of climatically sensitive lithologies.

**Climatically sensitive lithologies**. Climatically informative lithologies in the early Cambrian span all major Palaeozoic continents and equatorial to polar palaeolatitudes (Figs. 2, 3). In detail, lithology data coverage is uneven: there is a particular dearth in present-day South America, central Africa, Antarctica, and the Baltic region, whilst the rest of Europe, North America, and China are well covered. The majority of lithologies in the database are evaporitic; occurrences of glaciogenic and lateritic deposits are scarce. Lateritic deposits, characteristic of hot humid climatic conditions[12], are readily lost to post-burial diagenesis and weathering processes, so the dearth of early Cambrian laterites may be explained by their low preservation potential. The scarcity of glaciogenic deposits is less likely to be explained by low preservation potential and their virtual absence is more likely to be a climatic signal. It is notable that evaporite and oolite deposits occur at high (> 60°) latitudes in all configurations except configuration B. Configuration B is also the only reconstruction

under which there are obvious palaeo-latitudinal patterns in the distribution of lithologies in our dataset (Fig. 3), though this may simply reflect the concentration of continents in a narrower latitudinal belt under this configuration (Fig. 1).

**Global palaeoclimate simulations**. To explore plausible climatic conditions for each of the continental configurations, we produced GCM simulations at six atmospheric $p\mathrm{CO}_2$ levels spanning the range of values supported by long-term carbon cycle models ($p\mathrm{CO}_2 = 4, 8, 16, 32, 64,$ and 128 times PAL). We find that atmospheric $p\mathrm{CO}_2$ levels exert a stronger control on simulated sea-surface temperatures (SSTs) than continental configuration in our simulations. The lowest $p\mathrm{CO}_2$ simulations (4 PAL) have the coldest SSTs and widespread sea ice (Fig. 4; Supplementary Figs. 18 and 19). The 8 PAL $p\mathrm{CO}_2$ runs simulate SSTs comparable to the modern ocean, accompanied by substantial high-latitude sea ice in configurations A and C. The intermediate $p\mathrm{CO}_2$ runs (16 and 32 PAL) simulate SSTs exceeding modern values but within the tolerance range of modern animals (SST < 41 °C for subtidal ectothermal animals[50]) and limited to no sea ice (Fig. 4). The high $p\mathrm{CO}_2$ (64 and 128 PAL) runs simulate SSTs exceeding the tolerance range of modern animals from equatorial to mid-latitudes.

Palaeogeography exerts a systematic control on simulated climate: configurations B and C consistently lead to colder temperatures than A and D (Supplementary Fig. 7), with configuration C consistently having the coldest global temperatures under all boundary conditions. These trends mainly reflect differences in global ocean area, which is lower in configuration C (420 Mkm², 1 Mkm² = $10^6$ km²) than in configurations A (444 Mkm²), B (436 Mkm²), and D (448 Mkm²; Supplementary Fig. 22). Simulated temperatures increase with increasing ocean area, due to the relatively lower albedo of oceans compared to land. The simulated control of land–sea distribution on global climate is in line with previous studies conducted using different climate models on various Phanerozoic time slices[51,52].

To summarise the modelled climatic conditions, we applied the Köppen–Geiger climate classification scheme (see Methods). The distribution of modelled Köppen–Geiger climate classes is approximately zonal for all four configurations (Fig. 5). Tropical conditions are restricted to a narrow equatorial latitudinal belt (approximately ±30°) in the 4 PAL and 8 PAL $p\mathrm{CO}_2$ simulations, expand up to ±50° latitude by 32 PAL, and reach polar latitudes in the 128 PAL simulation (see Supplementary Figs. 23–32). A tropical monsoonal belt is simulated for approximately ±5°–30° latitude for all simulations. Arid conditions are almost entirely restricted to regions within approximately ±30° latitude in all simulation conditions, though there is some southwards expansion with increasing $p\mathrm{CO}_2$ of the continental arid region on Gondwana to approximately 45° S. The broad temperate climate classes are mostly fully humid, though all three types of temperate climate occur. Cold climate classes predominantly occur on high-latitude Gondwana in the Antarctocentric configurations, and also on parts of Baltica. Polar climates are predicted at high latitudes for all configurations at the lowest, 4 PAL, $p\mathrm{CO}_2$ simulations. In configurations B and C, polar conditions including significant build-ups of sea ice are found at mid- to low-latitudes. However, by 8 PAL $p\mathrm{CO}_2$ polar climates are only a minor component of simulations for all except configuration C, which consistently results in the coldest climates.

**Data–model comparison**. To test whether the distribution of climatically sensitive rock types is congruent with our climate model simulations for each of the four continental configurations, we calculated data–model agreement scores for 'lower Cambrian'

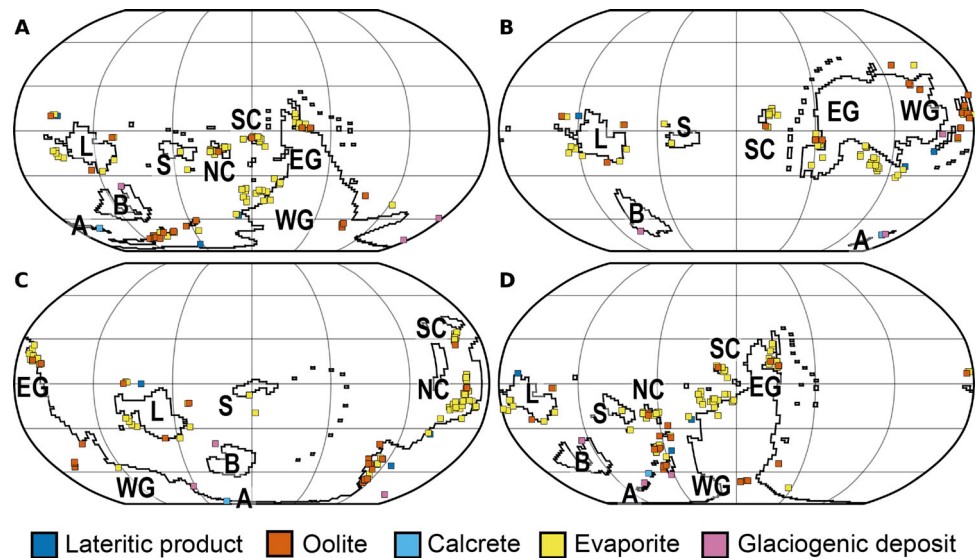

**Fig. 2 Palaeo-positions of lower Cambrian climatically sensitive lithologies plotted on each of the continental configurations (A–D) investigated here.** See Fig. 1 for explanation of the configurations and continent abbreviations.

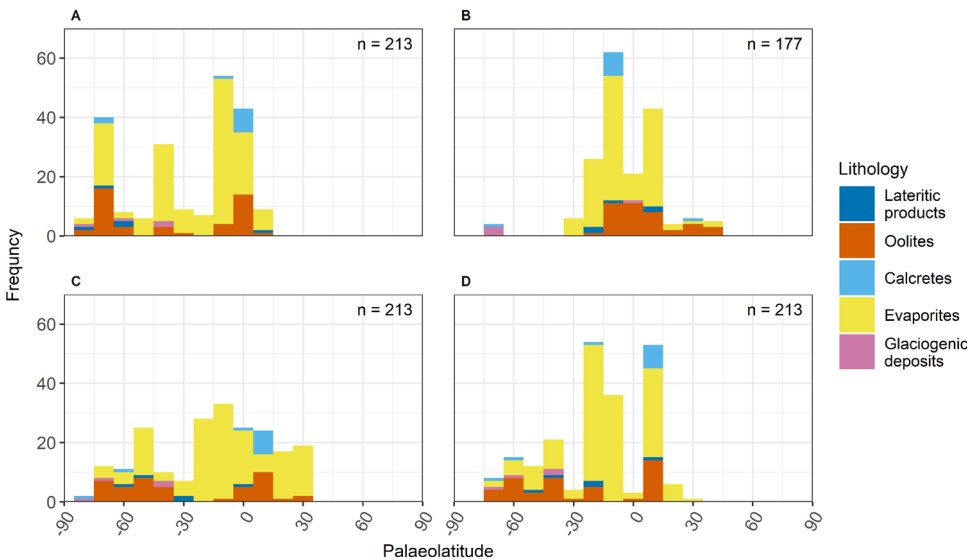

**Fig. 3 Zonal distribution of lower Cambrian climatically sensitive lithologies for each continental configuration (A–D).** Bin width = 10° latitude. See the Supplementary Information for 5° latitude bin width and breakdown by palaeocontinent. The sample size for configuration B is smaller due to the absence of North China in this configuration[18,19].

lithologies (Fig. 6; see 'Methods'). A score of one would indicate that the geological data and the model data are perfectly in concert; conversely, a score of zero would indicate that none of the geological data are in agreement with the modelled climate classes. Configuration B scores highest for all except the 4 PAL and 32 PAL $p$CO$_2$ simulations. Configuration D performs similarly well to configuration B for the 16, 32, and 64 PAL $p$CO$_2$ simulations. Configuration A performs weakly, and often weakest, for all simulations. Configuration C performs weakly for all but the lowest (4 PAL) and highest (128 PAL) $p$CO$_2$ simulations. Overall, under intermediate $p$CO$_2$ forcing, configurations B and D score comparably, and score better than configurations A and C.

We tested the stratigraphic sensitivity of our analyses by computing data–model agreement scores for only Cambrian Series 2 geological data (a subset of the lower Cambrian dataset). The Cambrian Series 2 analyses scored slightly weaker than the

lower Cambrian analyses, but the overall patterns are robust across both subsets of lithology data, and the magnitude of difference is small (see Supplementary Information).

Additional simulations conducted at 32 PAL $p$CO$_2$ show that these trends are also robust to FOAM simulations under contrasting orbital configurations (see Supplementary Information). Similarly, sensitivity tests confirm that the agreement scores are robust to the effect of small uncertainties in the palaeo-positions of our lithology data (see 'Methods' and Supplementary Information). These observations are supported by Tukey honestly significant difference (HSD) tests of the 32 PAL $p$CO$_2$ simulations for all orbital parameters (Supplementary Table 8) under which the scores for configurations B and D are not significantly separated (95% confidence interval) with palaeogeographic uncertainty of up to 250 km, and configurations A and C are not significantly separated when accounting for palaeogeographic uncertainty of 200 km to at least 500 km. A similar but

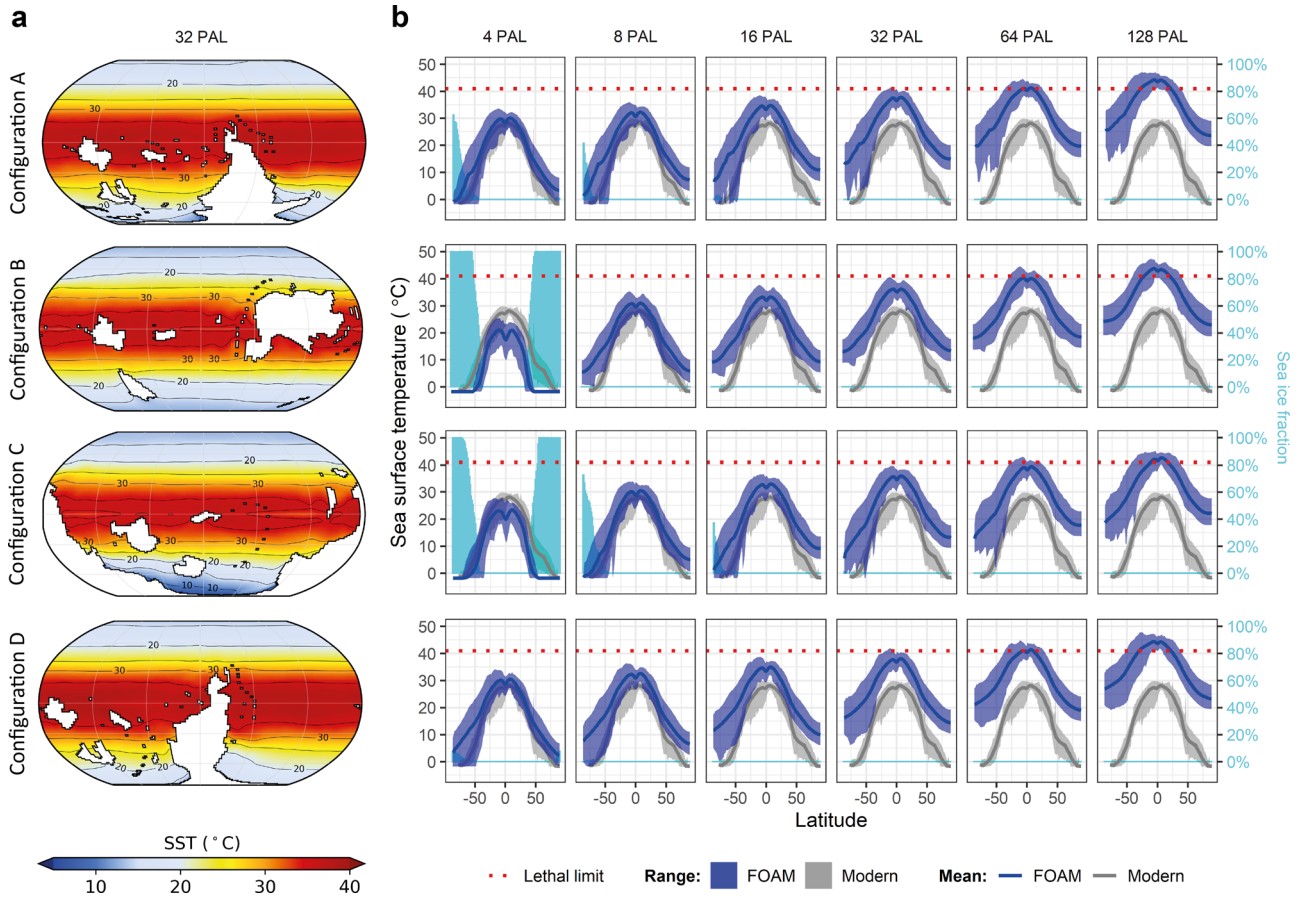

**Fig. 4 FOAM GCM simulations of early Cambrian sea-surface temperatures for each continental configuration (A–D) with present-day orbital parameters. a** Maps of simulated SST for configurations A–D at 32 PAL $pCO_2$. **b** Zonally averaged model (dark blue) SST annual mean and seasonal variation for each configuration and $pCO_2$ forcing, with comparable present-day SST values (grey) from ref. [72], and lethal temperature limit for animals of 41 °C, following ref. [50], plotted with mean zonal sea-ice cover (light blue). The sea ice cover represents the percentage of the ocean that is covered by sea ice (yearly average). $pCO_2$ levels relative to PAL (=280 ppm).

statistically weaker result is returned when comparing all simulations, including all $pCO_2$ levels, together (Supplementary Table 9).

In summary, equatorial configuration B performs well regardless of the GCM boundary conditions. The Antarctocentric configuration D performs similarly well with intermediate $pCO_2$ forcing conditions, but performs more weakly with the lower and higher $pCO_2$ forcings. Configurations B and D outperform A and C in our analyses, and we cannot robustly separate the configurations within these pairings.

## Discussion

Our GCM simulations support a greenhouse climate in the early Cambrian, consistent with previous qualitative interpretations[1,12,53,54]. Extensive sea ice is only simulated for all configurations at the lowest $pCO_2$ forcing (4 PAL), which is below $pCO_2$ levels indicated by long-term carbon cycle models for the early Cambrian[37,39–41]. The mid- to high-latitude polar conditions simulated in the 4 PAL $pCO_2$ runs may be expected to have left evidence of substantial glaciation in the rock record, robust evidence of which is not found[12]. In fact, detrital kaolinite, indicating hot humid conditions, is reported from the lower Cambrian of Algeria (northern Africa)[12,55], which is reconstructed at mid- to high-palaeolatitudes in the Antarctocentric configurations. Furthermore, Baltica is consistently reconstructed at mid- to high-palaeolatitudes in both Antarctocentric and equatorial configurations (Fig. 1) and has a well-developed lower

Cambrian stratigraphic record that lacks evidence for the extensive local glaciation[56] that is simulated at the low 4 PAL $pCO_2$ condition for configuration B (Fig. 4; Supplementary Fig. 23). The highest $pCO_2$ simulations (64 and 128 PAL) exceed the $pCO_2$ range supported by carbon cycle models (Supplementary Table 1) and also result in lethally high ocean temperatures[50,57] from the tropics to mid-latitudes (Fig. 4) whereas these environments are well-known to have been rich in marine animal life[1]. Therefore, the lowest (4 PAL) and highest (64 and 128 PAL) $pCO_2$ simulations may be rejected as unviable on grounds of climate and biology proxies, respectively.

We seek to further refine constraints on $pCO_2$ level and palaeogeography by examining the data–model agreement scores of the remaining $pCO_2$ simulations (8, 16, and 32 PAL). The overall patterns in agreement score, distinguishing configurations B and D with higher scores from configurations A and C with lower scores, are robust to small errors in palaeo-position of at least 250 km. Agreement scores for all simulations are less than 100%. This indicates that improvements are perhaps needed to the lithology database, the temporal resolution of our study, all of the tested early Cambrian palaeogeographic reconstructions, and/ or the modelling capabilities of FOAM. Nevertheless, a study of latest Cretaceous climate using a similar methodology found maximum "good agreement" between GCM and evaporite data of 86%[17]. This is comparable to the highest agreement scores in our analyses, which bolsters our confidence in the application of the lithology–GCM data comparison approach in deep time.

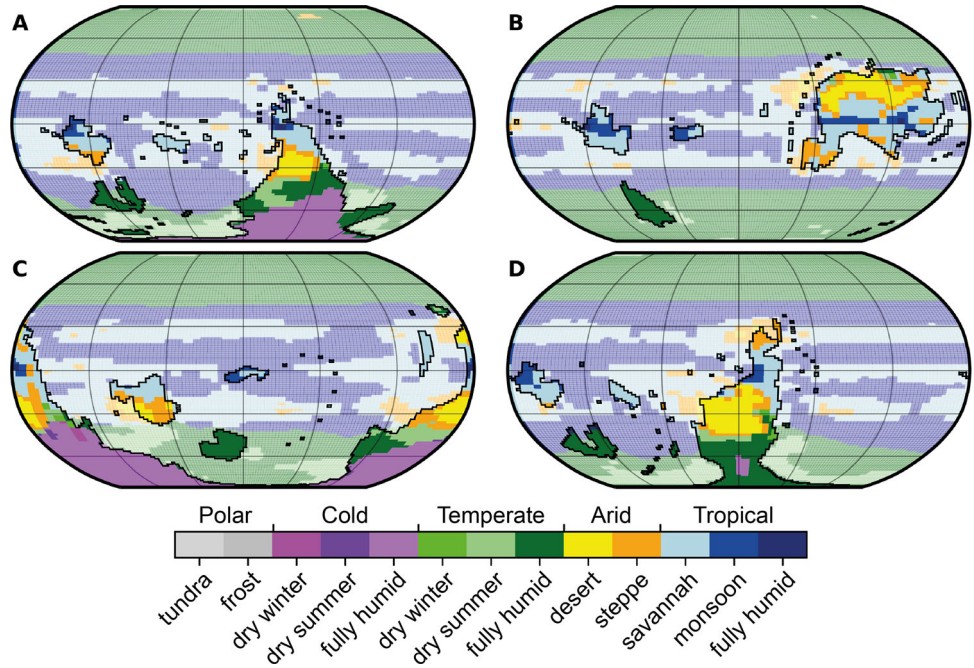

**Fig. 5 Simulated distributions of Köppen–Geiger climate classes for each configuration (A–D) simulated with 32 PAL $p$CO$_2$ and present-day orbital parameters.** Land areas are unfaded and outlined in black, ocean areas are faded. See 'Methods' and Supplementary Information for description and discussion of Köppen–Geiger climate classes.

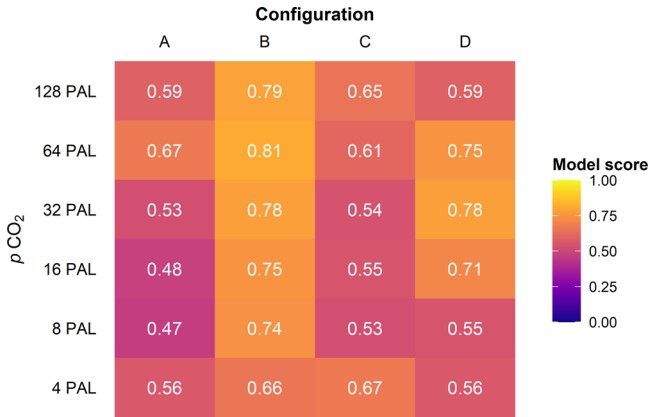

**Fig. 6 Data–model agreement scores for lower Cambrian lithology data and FOAM GCM simulations, incorporating palaeogeographic uncertainty of 200 km.** The scores compare Köppen–Geiger climate classes assigned to climatically sensitive rock types with the climate classes produced under each GCM simulation, following the agreement scoring equation of ref. [49] (see 'Methods'). Simulations were run with present-day orbital parameters.

Whilst the data–model agreement scores for configuration B are consistently high across all $p$CO$_2$ conditions, the highest-scoring Antarctocentric configuration (D) performs best in the highest of the intermediate $p$CO$_2$ simulations that were previously retained (i.e. at 32 PAL). This $p$CO$_2$ value is in line with the higher estimates of carbon cycle models (Supplementary Table 1)[37,39,41]. The SSTs simulated for configuration D at 32 PAL $p$CO$_2$ are also consistent with inferences from isotopic SST estimates for high-latitude Avalonia of approximately 20 °C–25 °C[7], while the 8 and 16 PAL $p$CO$_2$ runs simulate SSTs that are too cold. In configuration C, Avalonia is positioned at the South Pole and remains too cold under all except the 128 PAL

$p$CO$_2$ simulation. The isotopic temperatures of 30 °C to 41 °C suggested by ref. [8] for tropical Siberia, following different assumptions of seawater δ[18]O to ref. [7], are within the SST range supported by 8 PAL to 64 PAL simulations for all configurations.

Configuration B, the equatorial reconstruction tested here, was constructed primarily from the qualitative consideration of the distribution of lower Cambrian climatically sensitive lithologies[18,19]. The test we applied here is therefore somewhat circular for configuration B as it represents a qualitative expectation of the distribution of lower Cambrian climatically sensitive lithologies, and it is expected to perform strongly in our analyses, which it does. However, the question arises whether it outperforms Antarctocentric configurations which are constructed primarily from considerations of palaeobiogeographic and palaeomagnetic data[27,28,30,31]. The two Antarctocentric configurations most widely applied to understand early Cambrian palaeogeography tested here, A and C[25–27,30,31], perform weakest in our analyses. However, configuration D, a variant of the Antarctocentric paradigm informed by trilobite biogeography[28], performs as well as the equatorial configuration B in our 16 and 32 PAL $p$CO$_2$ analyses.

Therefore, a variant of the Antarctocentric palaeogeographic paradigm similar to configuration D, that is supported by palaeomagnetic and palaeobiogeographic data[28] and is consistent with succeeding well-constrained Ordovician plate tectonic reconstructions[27,31], can also explain the distribution of lower Cambrian climatically sensitive lithologies (Fig. 6). It is not necessary to invoke exceptional plate tectonic conditions to explain the palaeogeographic distribution of lower Cambrian climatically sensitive lithologies, contra ref. [18]. However, it is important to account for the distribution of climatically sensitive lithologies when reconstructing continental configurations in the early Palaeozoic, as demonstrated by the low data–model agreement scores of configurations A and C (Fig. 6). Our results highlight that a holistic approach, accounting for palaeomagnetic, palaeobiogeographic, and palaeoclimatic data, is required to

construct robust hypotheses of palaeogeography in deep time, and that confidence in these hypotheses can be quantitatively evaluated by comparison with GCM simulations. We suggest that this integrated approach will be particularly effective in other Palaeozoic and Proterozoic intervals where proxy data are sparse.

Our analyses suggest that the early Cambrian was likely characterised by a warm greenhouse climate. Because simulated temperatures are a function of the model climate sensitivity, we remind the reader that the numerical $pCO_2$ estimates are model-dependent. Because FOAM simulations are commonly slightly colder than those of more recent models for the same $pCO_2$ level, e.g. ref. [46], using a GCM of the latest generation would likely simulate similar climatic states with lower $pCO_2$ forcing. Our $pCO_2$ values can therefore be considered an upper estimate. Nevertheless, the magnitude of early Cambrian atmospheric $pCO_2$ levels predicted by long-term carbon cycle models [37,39–41] agrees with that suggested by our coupling of GCM and lithology data (16–32 PAL atmospheric $pCO_2$). Consistent with the general picture of a warm greenhouse climate, lithology data and GCM simulations do not support expansive or long-lived low-altitude glaciation in this interval. Antarctocentric early Cambrian continental configurations, which are supported by palaeomagnetic and palaeobiogeographic data [25–28,31], can be reconciled with the distributions of climatically sensitive lithologies. A palaeogeographic paradigm shift to an equatorial distribution of the continents is not required in the early Cambrian, but palaeoclimate and lithological data should be (semi-)quantitatively accounted for when reconstructing Cambrian palaeogeography. A deeper understanding of how the Earth System works in deep-time contexts such as the Cambrian, where geological data are sparse, can be achieved by an integrated approach combining numerical models and geological observations, underpinned by robust quantitative analyses.

## Methods
**Continental configurations**. We tested four hypothesised continental configurations for the Cambrian Period published in recent years (Fig. 1). Together, these configurations capture much of the range of palaeogeographic reconstructions used by palaeontologists studying the ecological and biogeographical patterns of the Cambrian animal radiation, though many other variants are available, e.g. refs. [21–24] and could be examined in future studies. The four configurations we tested are briefly described here.

*Configuration A* derives from the 510 Ma BugPlates[25] reconstruction, adapted with information from refs. [26,27]. This is primarily based on palaeomagnetic and brachiopod palaeobiogeographic data. BugPlates and related continental configurations have been widely used over the last decade by palaeontologists examining palaeobiogeographic patterns, e.g. ref. [58]. This is one of the many variants of the early Cambrian Antarctocentric palaeogeographic paradigm in which a portion of West Gondwana, in this case North Africa, lies over the South Pole.

*Configuration B* derives from the 'terminal Ediacaran–early Cambrian transition' reconstruction of refs. [18,19], which followed ref. [33]. This is one of the equatorial configurations that provide a striking alternative to the Antarctocentric paradigm[18]. The main difference between configuration B and the Antarctocentric paradigm is in the position of West Gondwana which lies at equatorial latitudes in this configuration. Configuration B is primarily based on lithostratigraphic relationships and inferences of climate from lithology, rather than palaeomagnetic or palaeobiogeographic data. It should also be noted that, without explanation, the North China palaeoplate does not appear on this configuration, despite the presence of a substantial succession of lower Cambrian rocks there, e.g. ref. [59]. In other configurations, North China is regarded as residing at low, equatorial to tropical, latitudes during the early Cambrian though its exact placement is problematic[23,26–28,30].

*Configuration C* derives from the 510 Ma PALEOMAP Project reconstruction[30,31]. Phanerozoic PALEOMAP reconstructions are associated with the GPlates software package and are widely used by the Palaeozoic palaeontological community, e.g. ref. [60]. Configuration C is an Antarctocentric reconstruction, broadly similar to A but with Gondwana rotated slightly anticlockwise, placing Avalonia rather than North Africa over the South Pole.

*Configuration D* is the reconstruction of ref. [28], which is an adaptation of a BugPlates[25] reconstruction augmented with trilobite palaeobiogeographic data. Configuration D is an Antarctocentric reconstruction in which Gondwana is

rotated clockwise with respect to A, so that northwestern South America resides over the South Pole, and North Africa spans mid- to high-palaeolatitudes.

**Climatically sensitive lithologies**. Climatically sensitive lithologies are rock types deposited under particular climatic conditions[12]. Each lithology may form under a range of climatic conditions, and non-climatic factors may intervene to prevent their deposition. Therefore, while the presence of a climatically sensitive lithology is evidence for the presence of particular climate conditions, or range of climate conditions, the absence of that lithology is not evidence of absence of the associated (range of) climate.

We compiled a database of climatically sensitive lithologies, augmenting the dataset of ref. [12] with additional data from published literature. Each entry in the analysed dataset includes at least one source reference, geographic location, stratigraphic information, and numerical depositional age range and best estimate (Supplementary Data 1). Numerical ages have been correlated to the Geologic Time Scale 2012[61].

Only some of the commonly considered climatically sensitive lithologies are available in the Cambrian; e.g. coal does not become a significant part of the rock record until the Devonian[12]. The key groups of lithologies used in this study are marine evaporites, calcretes and other duricrust deposits, lateritic deposits including bauxite, oolitic limestones, and glaciogenic sedimentary deposits (Table 1). We used literature data to determine the range of climatic conditions under which each lithology may form.

**GCM simulations**. We used the Fast Ocean Atmosphere Model (FOAM)[35] to simulate early Cambrian climate conditions. FOAM is a three-dimensional GCM with coupled ocean and atmosphere modules well-suited to tackling deep-time palaeoclimate questions[42–45]. The atmospheric module is an upgraded parallelised version of the National Center for Atmospheric Research (NCAR) Community Climate Model 2 (CCM2) which includes radiative and hydrologic physics from CCM3 version 3.2. The atmospheric module was run with R15 spectral resolution ($4.5° \times 7.5°$) and 18 vertical levels. The ocean module is Ocean Model version 3 (OM3) which has $2.8° \times 1.4°$ longitude–latitude resolution and 24 unevenly spaced vertical levels including the uppermost 0 m to 20 m and 20 m to *circa* 40 m surface ocean levels. There are no flux corrections in the coupled model. The model simulations were integrated for 2000–3000 years to reach deep-ocean equilibrium, the short turnaround time allowing for millennial-scale integrations. The climatology files used in these analyses were built from the last 50 years of the model runs (Supplementary Data 2: https://doi.org/10.5281/zenodo.4506617).

Due to the absence of vascular land plants in the Cambrian, e.g. ref. [62], the land surface was defined as a rocky desert with an albedo of 0.24, modified by snow if present. All simulations were initialized with a warm ice-free ocean of homogenous salinity (35‰). We used an early Cambrian solar luminosity of 1309.5 Wm$^{-2}$, 4.27% weaker than present day, following ref. [63]. To a first approximation, following the energy balance equation of ref. [64], an atmospheric $pCO_2$ value of about 10 PAL is required to compensate reduced solar luminosity in the Cambrian.

There are currently no reliable and precise $pCO_2$ proxy data for the Cambrian Period[36–38]. Long-term carbon cycle models are the best method currently available for constraining the early Phanerozoic atmospheric composition. Consequently, we examined a range of atmospheric $pCO_2$ values: 4, 8, 16, 32, 64, and 128 times preindustrial atmospheric levels (PAL = 280 ppm) $pCO_2$, following model constraints[37,39–41] (see Supplementary Table 1). The highest $pCO_2$ value that we employ is comfortably within the range for which the radiation code of FOAM is validated[42]. Note that, as in any modelling study, the $pCO_2$-temperature relationship is dependent on the model climate sensitivity. Using a GCM with a different climate sensitivity would result in support for a different range of numerical $pCO_2$ values.

Precise calculation of Earth's orbital conditions beyond a few tens of millions of years is not feasible, e.g. ref. [65]. We ran simulations for all $pCO_2$ levels with present-day orbital parameters. We tested the sensitivity of our analyses to changing orbital parameters by running simulations at 32 PAL $pCO_2$ for hot and cold austral summer, and high and low obliquity orbital parameters (Supplementary Table 2).

Simulations were run on the four selected continental configurations (A–D; Fig. 1). For configurations A, B, and D, topography and bathymetry were reconstructed using published geological data[66]. Six broad altitude categories were used (Supplementary Table 3), following the method previously adopted by ref. [67]. In the absence of better constraints, abyssal ocean plains were modelled as flat-bottomed, following previous Palaeozoic climate modelling studies, e.g. ref. [44]. Topographic and bathymetric data were provided with configuration C[30,31].

**Köppen–Geiger climate classification**. The Köppen–Geiger climate classification scheme is based on annual and seasonal thresholds in temperature and precipitation and comprises five primary classes: (A) tropical, (B) arid, (C) temperate, (D) cold, and (E) polar (see Supplementary Table 5)[47,48,68]. Four of the five primary classes (A, C, D, E) are defined by temperature thresholds. The arid class (B) is defined by the relationship of relating mean annual precipitation to a threshold temperature. The main climate classes are subdivided by precipitation and temperature thresholds. The polar climate class, E, is subdivided by temperature alone.

There is a third layer of subdivision for classes B, C, and D primarily determined by temperature seasonality, which is beyond the temporal resolution of our proxy data. In this study, we use only the first two layers of the Köppen–Geiger climate classification (see Supplementary Table 5 and Supplementary Data 3).

The Köppen–Geiger classification scheme was developed for the modern world, with a vegetated terrestrial realm, and the thresholds, therefore, reflect climatic factors limiting vegetation growth, like frost and drought. The lack of terrestrial vegetation during the early Cambrian[62] means that the precise implications of this classification should not be directly compared to the modern world. However, the fundamental climatic controls on climatically sensitive lithologies, namely temperature and precipitation, are the same climatic controls on vegetation, and are the basis of the Köppen–Geiger climate classification scheme[47,48,68]. Furthermore, the Köppen–Geiger climate classification scheme has been applied with some success to climatically sensitive lithologies in the more recent stratigraphic record[69]. Therefore, the Köppen–Geiger classification provides a common medium in which to interpret lithology and climate model data. Modelled temperature and precipitation results were used to assign each FOAM grid cell a Köppen–Geiger class. Each category of lithology data was assigned to the (range of) Köppen–Geiger climate classes under which they may form based on published literature data (Table 1 and Supplementary Table 6). Each lithology occurrence may be in agreement with multiple climate classes.

**Data–Model comparison**. We calculated agreement between the climate classes indicated by the lithology data and the modelled classes, following Eq. (1)[49]. Perfect agreement means that all of the climate classes simulated in FOAM match the tolerable range of climates indicated by every lithology occurrence. Perfect disagreement means that none of the modelled climate classes match the tolerable range of climates indicated by lithology data. In particular, for each FOAM simulation, the data–model agreement score ($\Phi$) for $n$ observations of weight $W_i$ and agreement $A_i$ is calculated as:

$$\Phi = \frac{1}{2}\left(\frac{\sum_{i=1:n} A_i W_i}{\sum_{i=1:n} W_i} + 1\right) \tag{1}$$

Where model and lithology data agree, $A = 1$, and where model and lithology data disagree, $A = -1$. Weight ($W$) is assigned a value of $0 \leq W \leq 1$ for each lithology datum depending on the uncertainty around the positive identification of the lithology and the climatic conditions required for the lithology to form and is, by necessity, a qualitative assessment. The calculated score for each FOAM simulation, $\Phi$, is a value between 0 and 1, where 0 equals perfect disagreement and 1 equals perfect agreement between the lithology dataset and FOAM simulations. The R code used to calculate agreement scores is provided in Supplementary Data 4.

The problems inherent to international correlation of strata of this age, e.g. refs. [21,70,71] and the loose age constraints on some of our lithology data warn against analysing tightly defined time slices. We, therefore, consider series-level stratigraphic resolution appropriate. We analysed two subsets of the lithology database, selecting in separate analyses either (i) traditional 'lower Cambrian' deposits (approximately equivalent to the Terreneuvian Series and Series 2), or (ii) Cambrian Series 2 deposits. Inherent to this approach is the time-averaging of climatic data which reflect much more variability. The epoch/series level resolution of our study, therefore, enables us to examine overall climate state, but not shorter timescale climatic variability.

Without explanation, the North China palaeoplate was not included in configuration B[18,19], despite having an extensive lower Cambrian stratigraphic record, e.g. ref. [59]. We performed additional sensitivity analyses to examine the biasing effect of removing all North China geological data from the data–model agreement calculations. These sensitivity analyses show that, whilst systematic loss of geological data from one palaeocontinent changes the numerical scores, the overall patterns remain robust to this bias (Supplementary Fig. 48).

**Palaeogeographic uncertainty**. In calculating data–model agreement scores, we assume that (a) lithology palaeo-positions are all accurately reconstructed; (b) the continental configurations are entirely accurate; and (c) the continental configurations are accurate for and apply to the whole depositional period of the analysed data set (either 'lower Cambrian' or Cambrian Series 2). These assumptions come with substantial uncertainty, so we tested the impact on our analyses of small errors in the geographical alignment of lithology and model data. We calculated scores accounting for palaeogeographic uncertainty of a given radius about each lithology data point.

To calculate palaeogeographic uncertainty of radius X km: (1) all model cells within X km were found; (2) agreement was scored (−1 or +1) for each identified model cell; (3) a mean agreement value (−1 to +1) was calculated for each lithology datum at radius X km; (4) for each simulation, the data–model agreement score (0 to +1) accounting for palaeogeographic uncertainty at radius X km was calculated using the mean agreement values from step (3), following the equation of ref. [49]; (5) The palaeogeographic uncertainty (0 to +1) at radius X km was calculated as the magnitude of the difference between the exact and uncertainty-adjusted scores for each simulation.

Scores that include palaeogeographic uncertainty (step 4) in their value are plotted without error bars. Scores that are calculated for only exact palaeo-

positions may be plotted with error bars (step 5) to show palaeogeographic uncertainty at a given radius. Palaeogeographic uncertainty scores were calculated for radii of 200 km, 250 km, 500 km, and 1000 km. The mean number of grid cells within each radius for each configuration are presented in Supplementary Table 7. Because model grid cells are determined by evenly spaced longitude and latitude lines in FOAM[35] they do not represent equal areas, and there will be more grid cells per unit radius at higher latitudes than at lower latitudes.

**Statistical tests**. We evaluated the differences between the data–model comparison scores for each configuration using two-way analysis of variance (ANOVA) and Tukey honestly significant difference (HSD) post-hoc tests. The Tukey HSD post-hoc test examines the distribution significance identified by an ANOVA test by making pair-wise comparisons of each sample mean. We performed separate Tukey HSD post-hoc tests on the 'lower Cambrian' and Cambrian Series 2 scores (Supplementary Tables 8 and 9). We performed Tukey HSD tests on agreement scores at each palaeogeographic uncertainty level. We performed Tukey HSD tests on the 32 PAL $pCO_2$ simulation scores to determine whether palaeogeographic differences were more significant than orbital configurations (Supplementary Table 8). We performed tests on all simulation scores (all $pCO_2$ forcings and orbital parameters), to determine whether palaeogeography overrides all other factors (Supplementary Table 9).

## Data availability
All data used in this paper are available on Zenodo. The geological data used in this manuscript are available as Excel and .csv files in the supplement to this paper. The analyses were conducted on the .csv file. The FOAM climate model outputs (NetCDF files) are available on the Zenodo repository.

## Code availability
The R scripts necessary to convert NetCDF files to Köppen–Geiger climate classes is available in the supplement to this manuscript (Supplementary Code 1). The R script necessary to compute agreement scores between geological and climate model data is available in the supplement to this manuscript (Supplementary Code 2). Both are also hosted on Zenodo.

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

## Acknowledgements

This project began during TWWH's PhD research, funded by NERC studentship NE/L0022493/1 within the Central England NERC Training Alliance (CENTA) with CASE partnership funding from the BGS (BUFI S266). It was finalised during TWWH's Ghent University Special Research Fund (BOF) Fellowship 01P12419. This project has received funding from the European Union's Horizon 2020 research and innovation programme under the Marie Sklodowska-Curie grant agreement No. 838373 to AP. Calculations were performed using HPC resources from DNUM CCUB (Centre de Calcul de l'Université de Bourgogne). TRAV was supported by Ghent University BOF grant BOF17/STA/013.

## Author contributions

T.W.W.H., A.P., and M.W. contributed equally to this work. T.W.W.H., A.P., and M.W. designed the study, with input from T.H.P.H. and Y.D. C.R.S. contributed data for configuration C.A.P. performed the GCM simulations. T.W.W.H. assembled the lithology data. T.W.W.H. prepared the R script and performed the analyses, with input from A.P., A.F., and P.S. T.W.W.H., A.P., and M.W. interpreted the results and wrote the first draft of the manuscript. All authors contributed to the discussion of results, and drafting and revising of the manuscript.

## Competing interests

The authors declare no competing interests.
