## [Peer Review File · Nature Communications]

REVIEWER COMMENTS

Reviewer #1 (Remarks to the Author):

This paper tackles an interesting problem: the nature of the Earth system during the Cambrian radiation of animals. The paper is generally well written and illustrated, and mostly accessible for a general audience. The article is not quite as thought-provoking as I had hoped after reading the title, but this can be easily fixed.

In the abstract, the motivation for the paper is given as Earth's environmental parameters in this interval being poorly constrained. I don't doubt that this is so, but it's a very general statement. I would expect some clear hypotheses to be briefly spelled out here that are being tested in the paper – including extremely different continental configurations, leading to different interpretations of climate proxies (such as lithologies) and different inferences about atmospheric CO₂. I think the motivation for the paper can be sharpened substantially. At the moment, the abstract merely states that four different continental configurations are being tested, ie as part of the methodology, but this is not cast in terms of Earth System hypothesis testing.

Climate sensitive lithologies are used as "ground-truth" data, but they are not all equally well explained. On line 134 it is explained that lateritic deposits are characteristic of hot humid climatic conditions, but there is no equivalent explanation in the main text for how other deposits are related to different climate belts, e.g. many readers may not know under which climate conditions oolites or calcretes form. The reader has to refer to the methodology to even learn that bauxite and oolites are all part of lateritic deposits. However, then, it becomes obvious that laterite and oolites are handled separately in the analysis, raising the question: what's different between them if oolites are just a type of lateritic deposit? Considering what an important role these data form in testing the climate models, this has to be made clearer in the main text of the paper, ie how exactly do the different indicators relate to the Köppen-Geiger climate belts?

Next, in Line 231, the "Tukey Honestly Significant Difference (HSD) test" shows up, without a reference, and without any explanation of what it actually is. I have never heard of it, and my guess would be that the same would hold for the vast majority of the readers of this paper. An explanatory sentence is needed here, explaining what it is, and how it works.

The discussion of the paper needs some extra work. The tests carried out by the authors are fascinating because they lead to lessons for how and how not to construct robust continental plate reconstructions in deep time. This is important because there tend to be "camps" of people doing reconstructions who either believe in the superiority of paleomag data or in that of geological data, including lithologies. The clear message indirectly contained in this paper is that palaeomagnetic and palaeobiogeographic data need to be combined to construct robust reconstructions, but this is not spelled out.

Lastly, I find the concluding sentence a little weak, i.e. that the approach may be used to more precisely test hypotheses about palaeogeography and palaeoclimate in other deep-time intervals, especially where quantitative proxy data are scarce. "hypotheses about palaeogeography and palaeoclimate"? Sounds pretty vague to me. The title of the paper starts with "Earth System boundary conditions". The end game is to understand how the Earth system works, right? It is also important to re-state that statistical evaluations of model outputs versus observations are really important. I think the outlook should be framed this way, ie that stating something along the lines that understanding how the Earth Systems works in deep geological time, where data are sparse, requires an inspired combination of numerical models and geological observations, underpinned by robust statistical methods. Can any more concrete examples be given, ie times of major change in the past, perhaps extinctions or major radiations, where this approach could be useful?

In summary, my recommended changes are all extremely minor, and I recommend that the paper largely be published as is. It was fun to read.

Reviewer #2 (Remarks to the Author):

The manuscript by Wong Hearing and co-authors employs lithologic climate indicators and climate model output converted to climate classes using the Koppen-Geiger classification to evaluate early Cambrian paleogeographic configurations and atmospheric pCO₂ estimates. On the basis of data-model agreement scores, the study determines that two paleogeographic reconstructions (B and D) best fit existing lithologic constraints and argue that early Cambrian atmospheric pCO₂ levels were between 16 to 32 PAL CO₂.

The manuscript has many strengths: It reminds us that lithologic indicators can provide useful paleoclimate—and paleogeographic—information. It includes an analysis, using data-model agreement scores, that is interesting and novel. It is very well written and illustrated. (Thanks to the authors for that.) For these reasons, I would support publication of the manuscript. Unfortunately, however, I am not confident for a couple of reasons that the conclusions drawn by the study are correct.

First, I find it disconcerting and unconvincing that the method would score B and D—too very different reconstructions—high, while scoring A and C—which are both generally similar to D—low. This conclusion doesn't pass common sense, and must indicate that the available lithologic indicators don't provide sufficient constraint on the climate regimes or that the position of one or two continental blocks is biasing the scores. The fact that reconstruction B places glaciogenic deposits at or near the equator and yet is scored high is further reason for concern that the lithologic indicators don't provide sufficient constraint.

Second, I am not convinced that the FOAM climate model is up to the task asked of it. The global average temperatures reported in this study under extremely high CO₂ levels are relatively low, even considering the reduced solar luminosity. FOAM is a couple of generations behind the very latest generation of climate models. The older generations of climate models have a low climate sensitivity and have historically not been able to simulate past warm climates (Eocene, Cretaceous) with proxy CO₂ values. The newest generation of climate models, largely due to their treatment of clouds, has a higher climate sensitivity. See, for example, Zhu et al., *Sciences Advances*, 2019. The low climate sensitivity in FOAM would affect the climate regimes estimated using the Koppen-Geiger classification and the conclusions regarding early Cambrian CO₂ levels.

L. 1-2. The title is catchy but does not describe very well the purpose of the study. It would lead one to believe that the paper discusses the early radiation of animals in a meaningful way, which it does not.

L. 58-59. "Coupling climatically sensitive lithologies and climate model simulations provides, for the first time..." This claim might be true for the early Cambrian; however, coupling climatically sensitive lithologies and climate model simulations is not new and was done fairly extensively in the 1990s. See, for example, Gibbs et al., *The Journal of Geology*, 2002. The authors should acknowledge this history, and also that lithologic indicators have been used to validate paleogeographic reconstructions as far back as Wegener.

Fig. 3. This is a nice figure. Why is the sample size in B smaller than in A, C, and D? It might be worth adding the reason to the figure caption.

Figure S7. The results in this figure are interesting. They show that the climate sensitivity in FOAM is linear, which is different than newer models (see Caballero and Huber, *Proc. Natl. Acad. Sci.*, 2013; Zhu et al., *Science Advances*, 2019), and should be commented on in the Discussion. They also show that the global temperature has a very low sensitivity to paleogeography. This raises the question of whether it was necessary to run the four simulations with different reconstructions. How different would the results have been if an aquaplanet simulation had been used and compared with the lithologic indicators in different

positions according to the four paleogeographic scenarios? Another way of answering this is to make a plot like Fig. 3 showing the distribution of Koppen-Geiger climate regimes using model output (instead of lithologic indicators). This would make a nice addition to the paper.

REVIEWER COMMENTS

Our responses are in black type.

Reviewer #1 (Remarks to the Author):

1. This paper tackles an interesting problem: the nature of the Earth system during the Cambrian radiation of animals. The paper is generally well written and illustrated, and mostly accessible for a general audience. The article is not quite as thought-provoking as I had hoped after reading the title, but this can be easily fixed.

We are grateful to Reviewer #1 for their considered and constructive review. We have recast aspects of our manuscript in light of their review, as detailed below.

2. In the abstract, the motivation for the paper is given as Earth's environmental parameters in this interval being poorly constrained. I don't doubt that this is so, but it's a very general statement. I would expect some clear hypotheses to be briefly spelled out here that are being tested in the paper – including extremely different continental configurations, leading to different interpretations of climate proxies (such as lithologies) and different inferences about atmospheric CO₂. I think the motivation for the paper can be sharpened substantially. At the moment, the abstract merely states that four different continental configurations are being tested, ie as part of the methodology, but this is not cast in terms of Earth System hypothesis testing.

We have rewritten the abstract as suggested by the reviewer and are grateful for these comments which help strengthen the manuscript.

3. Climate sensitive lithologies are used as “ground-truth” data, but they are not all equally well explained.

We agree, and have moved:

- (a) some of the explanation from Methods to the main text Introduction, and
- (b) Table S1 to the Methods as Table 1.

4. On line 134 it is explained that lateritic deposits are characteristic of hot humid climatic conditions, but there is no equivalent explanation in the main text for how other deposits are related to different climate belts, e.g. many readers may not know under which climate conditions oolites or calcretes form. The reader has to refer to the methodology to even learn that bauxite and oolites are all part of lateritic deposits.

We agree that it would be useful to have more of this material in the main text. See point (3) above.

5. However, then, it becomes obvious that laterite and oolites are handled separately in the analysis, raising the question: what's different between them if oolites are just a type of lateritic deposit? Considering what an important role these data form in testing the climate models, this has to be made clearer in the main text of the paper, ie how exactly do the different indicators relate to the Köppen-Geiger climate belts?

We assumed that many readers would be familiar with the environmental significance of such deposits (or refer to the methodology where not), but the reviewer highlights a very important point that requires better clarity in the manuscript. See our response to point (3) above.

On the substantive example given, we think that Reviewer #1 has conflated oolitic ironstones, which are a type of lateritic deposit, and oolitic limestones, which are not. This conflation emphasizes the need to improve the clarity of this aspect of the manuscript, which we have now done.

6. *Next, in Line 231, the “Tukey Honestly Significant Difference (HSD) test” shows up, without a reference, and without any explanation of what it actually is. I have never heard of it, and my guess would be that the same would hold for the vast majority of the readers of this paper. An explanatory sentence is needed here, explaining what it is, and how it works.* The Tukey HSD test is a standard statistical method for testing whether two different samples are drawn from populations with different distributions (different means and standard deviations). It is commonly used without reference, but we appreciate that many readers of this manuscript may not be familiar with this test and we now provide a brief explanatory sentence in the “Statistical tests” section of Methods.

7. *The discussion of the paper needs some extra work. The tests carried out by the authors are fascinating because they lead to lessons for how and how not to construct robust continental plate reconstructions in deep time. This is important because there tend to be “camps” of people doing reconstructions who either believe in the superiority of paleomag data or in that of geological data, including lithologies. The clear message indirectly contained in this paper is that palaeomagnetic and palaeobiogeographic data need to be combined to construct robust reconstructions, but this is not spelled out.*

We agree that this is a major conclusion from the manuscript (that an integrative approach is best), and have amended the discussion to better reflect this.

8. *Lastly, I find the concluding sentence a little weak, i.e. that the approach may be used to more precisely test hypotheses about palaeogeography and palaeoclimate in other deep-time intervals, especially where quantitative proxy data are scarce. “hypotheses about palaeogeography and palaeoclimate”? Sounds pretty vague to me. The title of the paper starts with “Earth System boundary conditions”. The end game is to understand how the Earth system works, right? It is also important to re-state that statistical evaluations of model outputs versus observations are really important. I think the outlook should be framed this way, ie that stating something along the lines that understanding how the Earth Systems works in deep geological time, where data are sparse, requires an inspired combination of numerical models and geological observations, underpinned by robust statistical methods. Can any more concrete examples be given, ie times of major change in the past, perhaps extinctions or major radiations, where this approach could be useful?*

We thank Reviewer #1 for emphasising the need for the full importance of this paper to be stated more explicitly. We have recast aspects of the abstract, introduction and discussion with this in mind, including the closing statement, and we now make explicit some of the instances in which this method may be applied (e.g. “other Palaeozoic and Proterozoic intervals”).

9. *In summary, my recommended changes are all extremely minor, and I recommend that the paper largely be published as is. It was fun to read.*

We are grateful for the reviewer’s positive comments that have helped us to make the manuscript sharper and more insightful.

Reviewer #2 (Remarks to the Author):

10. The manuscript by Wong Hearing and co-authors employs lithologic climate indicators and climate model output converted to climate classes using the Koppen-Geiger classification to evaluate early Cambrian paleogeographic configurations and atmospheric pCO₂ estimates. On the basis of data-model agreement scores, the study determines that two paleogeographic reconstructions (B and D) best fit existing lithologic constraints and argue that early Cambrian atmospheric pCO₂ levels were between 16 to 32 PAL CO₂.

We would like to thank Reviewer #2 for their considered and constructive review.

11. The manuscript has many strengths: It reminds us that lithologic indicators can provide useful paleoclimate—and paleogeographic—information. It includes an analysis, using data-model agreement scores, that is interesting and novel. It is very well written and illustrated. (Thanks to the authors for that.) For these reasons, I would support publication of the manuscript. Unfortunately, however, I am not confident for a couple of reasons that the conclusions drawn by the study are correct.

We are grateful for the complimentary comments from Reviewer #2, and are glad that they recognise the interest and novelty of this study. We address their concerns as raised in detail below.

12. First, I find it disconcerting and unconvincing that the method would score B and D—too very different reconstructions—high, while scoring A and C—which are both generally similar to D—low.

The method we employ in this study scores agreement at a local level and so is dependent on both regional climate and the position of every lithology data point in each continental configuration. Therefore, whilst there are similarities between the Antartocentric configurations A, C, and D, there are also important differences, in particular the rotation of Gondwana which places different regions at the South Pole and at mid-latitudes in each configuration.

13. This conclusion doesn't pass common sense, and must indicate that the available lithologic indicators don't provide sufficient constraint on the climate regimes or that the position of one or two continental blocks is biasing the scores.

We do not think this is the case. As described above (point 12), there are significant regional differences between the Antartocentric configurations, notably the rotation of Gondwana, which explain the differences in agreement score between these continental reconstructions. Indeed, this could be argued as “one or two continental blocks biasing the scores”, with the reconstructed rotation of Gondwana being of primary importance in reconstructing early Cambrian palaeogeography. However, this is not so much a bias as a primary result, and indicates that the North African part of Gondwana cannot be reconciled with a high palaeolatitudinal position. The fact that B scores well is expected; the fact that D also scores well indicates that there are non-unique solutions from this method: we must and do look elsewhere (palaeomagnetic and biogeographic data) for more information. This method is only one part of the solution, but it is an important part.

14. The fact that reconstruction B places glaciogenic deposits at or near the equator and yet is scored high is further reason for concern that the lithologic indicators don't provide sufficient constraint.

Firstly, the calculated agreement scores are aggregate scores for each simulation with all of the geological data. Individual, seemingly anomalous, deposits will therefore not substantially alter the scores for each simulation. These anomalies may arise due to:

- (a) Uncertainty in interpretation of the geological deposit,
- (b) Uncertainty in the climatic conditions required for the geological deposit,
- (c) Short timescale climate variability and/or time-averaging of geological data,
- (d) Uncertainty in the depositional age of the geological deposit, or
- (e) Uncertainty in the palaeo-position of the deposit (including palaeo-altitude).

In this specific case, there is a single glaciogenic deposit at low palaeo-latitudes in configuration B – a tillite deposit (G005 in our Supplementary Data 1) from Algeria, North Africa (Bertrand-Sarfati *et al.* 1995; Chumakov 2007). This deposit is confidently considered glaciogenic, and is too extensive to be considered the result of short timescale variability inducing very localized equatorial glaciation (Bertrand-Sarfati *et al.* 1995; Chumakov 2007), negating options a-c above. Age uncertainty (option d) is a possible explanation for this seemingly anomalous deposit: whilst the most likely age of the Algerian tillite is considered to be early Cambrian, uncertainty in the dating evidence also allows a late Ediacaran age assignment (Bertrand-Sarfati *et al.* 1995). Considering palaeo-position (option e): in configuration B, North Africa is reconstructed at equatorial latitudes, whilst in the 'Antarctocentric' configurations North Africa is reconstructed at mid- to high-palaeo-latitudes. In our opinion, the main cause of this anomaly in configuration B is that the palaeo-position of North Africa is incorrect – it should be at higher palaeo-latitudes as it is in the Antarctic configurations in which deposit G005 occurs at >60 °S (Figure 2). A short-lived and/or local cold interval in the early Cambrian or late Ediacaran is more likely at such high latitudes than at the low latitude required by configuration B.

Although configuration B was reconstructed based on qualitative assessments of the expected distribution of climatically sensitive lithologies, this was not a comprehensive approach including all available sedimentological evidence. Our effort is more comprehensive in its data coverage, and this deposit provides a qualitative reason for treating configuration B with caution.

*15. Second, I am not convinced that the FOAM climate model is up to the task asked of it. The global average temperatures reported in this study under extremely high CO2 levels are relatively low, even considering the reduced solar luminosity. FOAM is a couple of generations behind the very latest generation of climate models. The older generations of climate models have a low climate sensitivity and have historically not been able to simulate past warm climates (Eocene, Cretaceous) with proxy CO2 values. The newest generation of climate models, largely due to their treatment of clouds, has a higher climate sensitivity. See, for example, Zhu *et al.*, *Sciences Advances*, 2019.*

Reviewer #2 suggests that the equilibrium climate sensitivity (ECS) of FOAM is too low, and that this biases the simulated climate states. **The quantitative analysis provided below shows that the ECS of FOAM is not a limitation to this work.**

Table R1: Global mean surface temperatures (GMSTs) and ECS from our FOAM simulations for configuration D with present-day orbital parameters. PAL = preindustrial atmospheric

level (1 PAL $\text{CO}_2 = 280 \text{ ppm}$).

CO_2 (PAL)	GMST ($^{\circ}\text{C}$)	ECS ($^{\circ}\text{C}$)
4	20.89	
8	23.44	2.55
16	26.09	2.65
32	29.58	3.49
64	33.67	4.09
128	37.93	4.26

First, the ECS of our FOAM simulations is within the range defined by CMIP5 and CMIP6 models (Fig. R1). Table R1 shows the ECS values computed based on the FOAM output produced for this study between 4 and 128 PAL for configuration D. The ECS calculated based on these simulations ranges from 2.55 $^{\circ}\text{C}$ (for 4 to 8 PAL) to 4.26 $^{\circ}\text{C}$ (for 64 to 128 PAL). The mean climatic sensitivity value calculated over the whole $p\text{CO}_2$ range investigated, between 4 and 128 PAL, amounts to 3.41 $^{\circ}\text{C}$. This climate sensitivity is comparable to ECS values simulated in previous studies that applied FOAM to other geological periods, such as the Ordovician and Cretaceous (Pohl et al. 2014, 2020).

Multimodel mean ECS values calculated by Zelinka et al. (2020) for the CMIP5 and CMIP6 ensembles amount to 3.3 $^{\circ}\text{C}$ and 3.9 $^{\circ}\text{C}$, respectively. FOAM's mean climatic sensitivity calculated over the whole $p\text{CO}_2$ range investigated in this study lies between these values (3.41 K for successive $p\text{CO}_2$ doublings across 4 to 128 PAL for configuration D; Fig. R1). In addition, the ECS values calculated for successive $p\text{CO}_2$ doublings for our simulations (Table R1; error bar in Fig. R1) are within the range defined by the various CMIP5 and CMIP6 models (Zelinka et al. 2020).

Figure R1: Comparison of the ECS of FOAM with the CMIP5 and CMIP6 models. Climatic sensitivity for each individual CMIP5 (unfilled blue circles) and CMIP6 (unfilled orange circles) model and respective multimodel averages (filled circles) after Zelinka et al. (2020). The orange star represents the climatic sensitivity of CESM2, after Zhu *et al.* (2020a), which is at the upper end of the CMIP6 distribution. Cambrian FOAM ECS after Table R1: the filled black circle represents the mean value calculated between 4 and 128 PAL $p\text{CO}_2$ for configuration D, the error bar represents the range of values calculated between subsequent $p\text{CO}_2$ doublings over the same range of $p\text{CO}_2$ values (see text and Table R1). Figure modified after Zelinka et al. (2020).

We agree with the reviewer that some CMIP5 and CMIP6 models are characterized by a significantly higher ECS than FOAM. However, this higher ECS is not necessarily an asset. Indeed, Zhu *et al.* (2020a) demonstrated that the very high ECS of the CMIP6 model CESM2 (5.3 °C, see orange star in Fig. R1) is not supported by Early Eocene Climate Optimum (EECO) proxy data comprising coupled $p\text{CO}_2$ proxies and ocean temperature estimates. The previous generation model CESM1, characterized by a lower ECS of 4.2 °C (which is at the upper end of the ECS for our Cambrian FOAM simulations; Fig. R1), aligns much better with EECO proxies (Zhu *et al.* 2020a, b).

Figure R2: $p\text{CO}_2$ estimates derived from the proxy compilation of Foster *et al.* (2017) highlighting 90 Ma.

Reviewer #2 states that models of previous generations, including FOAM, cannot simulate warm climates, like those of the Eocene and Cretaceous, in agreement with proxy data. We first emphasize that FOAM has been used to successfully simulate global climate during both periods of time, as testified by recent publications on the Eocene (Chaboureaux *et al.* 2014; Ladant *et al.* 2014; Licht *et al.* 2014) and Cretaceous (Donnadieu *et al.* 2016; Pohl *et al.* 2020). For the Eocene, in particular, FOAM has recently been shown to perform relatively well in the context of a model intercomparison project (Hutchinson *et al.* 2020).

In order to be more specific, we provide below a comparison of proxy data with simulation results from both FOAM and the French Earth System model used for CMIP5 (Dufresne *et al.* 2013; Sepulchre *et al.* 2020), for the Late Cretaceous (Cenomanian-Turonian, ca. 90 Ma). We extracted $p\text{CO}_2$ estimates for 90 Ma from the recent proxy compilation of Foster *et al.* (2017): the best-guess value from the LOESS fit, and the minimum and maximum estimates within 68 % confidence (min = 136 ppm, best guess = 450 ppm, and max = 1120 ppm; Fig. R2). Then, we selected the FOAM simulations of Pohl *et al.* (2020) for 90 Ma, which best fit the three $p\text{CO}_2$ estimates: 140, 420 and 1120 ppm, respectively. We compared the ocean temperatures simulated in these three FOAM outputs with the

temperature estimates compiled by Laugié et al. (2020) for the Cenomanian-Turonian (Fig. R3). Model-data alignment improves monotonically between 140 and 1120 ppm (Fig. R3A-C). At 1120 ppm, model and proxy temperature estimates agree very well. This shows that the FOAM model is capable of simulating the climatic state of the Cretaceous thermal optimum when using $p\text{CO}_2$ levels supported by proxy data. We admit that best model-data match is obtained when using the upper-bound $p\text{CO}_2$ value of Foster et al. 2017 (within 68 % confidence). However, more recent climate models would not necessarily lead to different conclusions. Laugié et al. (2020), using the CMIP5 model IPSL-CM5A2, also found best model-data agreement for a $p\text{CO}_2$ level of 4 PAL (1120 ppm; Fig. R3D). The model-data comparisons for FOAM and IPSL-CM5A2 at 4 PAL are very similar, as shown in Fig. R3C and Fig. R3D.

Figure R3: Comparison between simulated sea surface temperatures and proxy data for the Cenomanian-Turonian. Proxy temperature estimates are from the Cenomanian-Turonian compilation of Laugié et al. (2020). (A-C): Results of FOAM experiments conducted on the continental reconstruction of Scotese and Wright (2018) for 90 Ma (Pohl *et al.* 2020) under three $p\text{CO}_2$ values supported by the $p\text{CO}_2$ proxy compilation of Foster et al. (2017): 140 ppm (panel A; lower bound of the 68 % confidence interval in Fig. R2), 420 ppm (panel B; LOESS fit in Fig. R2), and 1120 ppm (panel C; upper bound of the 68 % confidence interval in Fig. R2). (D) CMIP5 model IPSL-CM5A2 at 4 PAL (1120 ppm; Laugié et al. 2020), the same $p\text{CO}_2$ level as (C). The solid lines correspond to modelled mean annual temperatures, dashed lines correspond to winter and summer seasonal averages, and the grey shaded areas correspond to the annual temperature range.

In summary, the FOAM ECS matches the multimodel mean ECS values calculated by Zelinka et al. (2020) for the CMIP5 and CMIP6 ensembles, while contrasting with the excessively high ECS value of CESM2, which is rejected by means of comparison with proxy data (Zhu *et al.* 2020a). We also demonstrated that FOAM is capable of simulating past warm climates in

agreement with most recent $p\text{CO}_2$ proxy compilations and leads to results that are very similar to the ones provided by the CMIP5 model IPSL-CM5A2. **For these reasons, we think that (1) the use of FOAM is not a critical limitation to our study, (2) that our conclusions are robust, and (3) that using another model would not necessarily improve our work.**

In addition to the scientific arguments provided above against the replacement of FOAM by another model, there are also practical reasons against using most CMIP6 models:

- **The computational outlay associated with the use of recent CMIP models would not permit us to conduct the sensitivity analysis** (to orbital parameters and $p\text{CO}_2$, for a total of 40 simulations) provided in this work, which is crucial when working so deep in time due to the uncertainty in the model boundary conditions. In addition, we keep in mind that the use of CESM2, one of the CMIP6 models that perform best in the present day (Zhu *et al.* 2020a), would probably weaken our study due to the biases induced by the very high ECS of the model (Zhu *et al.* 2020a).
- **Most CMIP5 and CMIP6 models, in addition, are not flexible enough to permit the use of a Cambrian continental reconstruction.** The French CMIP5 model IPSL-CM5A2, for instance, uses a tripolar ocean grid. Each singular point of the ocean grid (i.e., the 3 poles) has to be placed on land. This limitation prohibits the use of the model beyond ca. 140 Ma.

Finally, we want to emphasize that **the FOAM model is still routinely used, with success, to simulate deep-time climates.** To illustrate this, we provide below a (non-exhaustive) list of recent publications that used FOAM:

- Botsyun, S., Sepulchre, P., Donnadieu, Y., Risi, C., Licht, A., Caves Rugenstein, J.K., 2019. Revised paleoaltimetry data show low Tibetan Plateau elevation during the Eocene. *Science* 363, eaaq1436.
- Donnadieu, Y., Puc at, E., Moiroud, M., Guillocheau, F., Deconinck, J.F., 2016. A better-ventilated ocean triggered by Late Cretaceous changes in continental configuration. *Nat. Commun.* 7, 10316.
- Hearing, T.W., Harvey, T.H.P., Williams, M., Leng, M.J., Lamb, A.L., Wilby, P.R., Gabbott, S.E., Pohl, A., Donnadieu, Y., 2018. An early Cambrian greenhouse climate. *Sci. Adv.* 4, eaar5690.
- Ladant, J.-B., Donnadieu, Y., 2016. Palaeogeographic regulation of glacial events during the Cretaceous supergreenhouse. *Nat. Commun.* 7, 12771.
- Licht, A., van Cappelle, M., Abels, H.A., Ladant, J.B., Trabucho-Alexandre, J., France-Lanord, C., Donnadieu, Y., Vandenberghe, J., Rigaudier, T., L ecuyer, C., Terry D, J., Adriaens, R., Boura, A., Guo, Z., Soe, A.N., Quade, J., Dupont-Nivet, G., Jaeger, J.J., 2014. Asian monsoons in a late Eocene greenhouse world. *Nature* 513, 501–506.
- Pohl, A., Austermann, J., 2018. A sea-level fingerprint of the Late Ordovician ice-sheet collapse. *Geology* 46, 595–598.
- Pohl, A., Donnadieu, Y., Godderis, Y., Lanteaume, C., Hairabian, A., Frau, C., Michel, J., Laugi , M., Reijmer, J.J.G., Scotese, C.R., Jean, B., 2020. Carbonate platform production during the Cretaceous. *GSA Bull.* 132, 2606-2610.
- Pohl, A., Donnadieu, Y., Le Hir, G., Ladant, J.B., Dumas, C., Alvarez-Solas, J., Vandembroucke, T.R.A., 2016. Glacial onset predated Late Ordovician climate cooling. *Paleoceanography* 31, 800–821.

- Porada, P., Lenton, T.M., Pohl, A., Weber, B., Mander, L., Donnadieu, Y., Beer, C., Pöschl, U., Kleidon, A., 2016. High potential for weathering and climate effects of non-vascular vegetation in the Late Ordovician. *Nat. Commun.* 7, 12113.
- Ruvalcaba Baroni, I., Pohl, A., van Helmond, N.A.G.M., Papadomanolaki, N.M., Coe, A.L., Cohen, A.S., van de Schootbrugge, B., Donnadieu, Y., Slomp, C.P., 2018. Ocean Circulation in the Toarcian (Early Jurassic): A Key Control on Deoxygenation and Carbon Burial on the European Shelf. *Paleoceanogr. Paleoclimatology* 33, 994–1012.
- Saupe, E., Qiao, H., Donnadieu, Y., Farnsworth, A., Kennedy-Asser, A., Ladant, J.-B., Lunt, D., Pohl, A., Valdes, P., Finnegan, S., 2019. Extinction intensity during Ordovician and Cenozoic glaciations explained by cooling and palaeogeography. *Nat. Geosci.* 13, 65–70.

In conclusion, we think that FOAM is well suited for the present contribution and did not repeat all 40 experiments using another model. However, we agree with the reviewer that uncertainties in ECS values, as in any modelling study, may impact the CO₂-temperature relationship and lead us to support slightly different CO₂ values if we used another general circulation model. In the revised version of the manuscript, we now clearly identify this limitation on lines 416–420.

16. The low climate sensitivity in FOAM would affect the climate regimes estimated using the Koppen-Geiger classification and the conclusions regarding early Cambrian CO₂ levels. We have demonstrated that the FOAM climate sensitivity is no limitation to our study (see above).

17. L. 1-2. The title is catchy but does not describe very well the purpose of the study. It would lead one to believe that the paper discusses the early radiation of animals in a meaningful way, which it does not.

We disagree. “Earth System boundary conditions during the early Cambrian radiation of animals” indicates that we are trying to constrain Earth System boundary conditions at the time of the early Cambrian radiation of animals. It does not imply that we are discussing the Cambrian radiation which is well handled elsewhere in the literature.

18. L. 58-59. “Coupling climatically sensitive lithologies and climate model simulations provides, for the first time...” This claim might be true for the early Cambrian; however, coupling climatically sensitive lithologies and climate model simulations is not new and was done fairly extensively in the 1990s. See, for example, Gibbs et al., The Journal of Geology, 2002. The authors should acknowledge this history, and also that lithologic indicators have been used to validate paleogeographic reconstructions as far back as Wegener.

Reviewer #2 is entirely right to highlight this history, but we do not consider their criticism to be entirely fair in this instance. The full quote from our original manuscript, lines 56-60, read:

“Climatically sensitive lithologies have been used with some success to reconstruct Earth’s climate throughout the Phanerozoic^{e.g. 12}, and can be used to evaluate the predictions of climate models¹³⁻¹⁶. Coupling climatically sensitive lithologies and climate model simulations provides, for the first time, a robust first-order method to reconstruct early Cambrian climate conditions on a global scale.”

References 13-16 are previous studies that do just this, spanning 1992 to 2012. Our study is the first to apply this method to Cambrian deposits. Gibbs et al. (2002) is another excellent example of this type, and we are content to cite this Palaeozoic example as well. For clarity, we have rephrased the concluding sentence of this paragraph to read: *“Coupling climatically sensitive lithologies and climate model simulations provides, for the first time in the early Cambrian, a robust first-order method to reconstruct climate conditions on a global scale.”*

19. Fig. 3. This is a nice figure. Why is the sample size in B smaller than in A, C, and D? It might be worth adding the reason to the figure caption.

Configuration B inexplicably omits the North China palaeocontinent (see Landing et al. 2013a, b), so those data are not included in the data-model agreement scores for B. We have added an explanatory note to the figure caption.

20. Figure S7. The results in this figure are interesting. They show that the climate sensitivity in FOAM is linear, which is different than newer models (see Caballero and Huber, Proc. Natl. Acad. Sci., 2013; Zhu et al., Science Advances, 2019), and should be commented on in the Discussion.

Actually, the ECS of FOAM is not linear. This is difficult to see in Fig. S7, which shows the results of many model runs, but Table R1 and associated Fig. R4 confirm that the FOAM ECS does increase with global temperature.

Figure R4: Equilibrium climate sensitivity (ECS) as a function of pCO₂. Values after Table R1.

21. They also show that the global temperature has a very low sensitivity to paleogeography.

We do not think that this statement is correct. Figure S22 (copied below) shows a systematic impact of the continental reconstruction on simulated mean global surface temperatures, which is commented upon in the main text on lines 177–185 (172–180 in the original manuscript; copied below as well). More importantly, we emphasize that regional climate changes are of primary importance in the present study, since they drive the agreement score calculated between models and proxy data and, consequently, our conclusions. Two continental configurations may well be characterized by very similar global temperatures and show very different agreements with the geological database due to regional differences in the position of the landmasses and, thus, simulated climate. The detailed analysis of the

capability of each continental reconstruction to capture (or not) the database, which is evaluated using a sophisticated scoring method, contrasts with previous studies and constitutes the core and value of our work.

172 Palaeogeography exerts a systematic control on simulated climate: configurations B
 173 and C consistently lead to colder temperatures than A and D (Figure S7), with configuration C
 174 consistently having the coldest global temperatures under all boundary conditions. These
 175 trends mainly reflect differences in global ocean area, which is lower in configuration C (420
 176 Mkm², 1 Mkm² = 10⁶ km²) than in configurations A (444 Mkm²), B (436 Mkm²), and D (448
 177 Mkm²; Figure S22). Simulated temperatures increase with increasing ocean area, due to the
 178 relatively lower albedo of oceans compared to land. The simulated control of land-sea
 179 distribution on global climate is in line with previous studies conducted using different
 180 climate models on various Phanerozoic time slices^{46,47}.

176
 177 Figure S22. Mean annual globally averaged modelled surface air temperatures at 16 PAL
 178 (blue), 32 PAL (black) and 64 PAL (red) as a function of the total ocean area for continental
 179 configurations A to D. Each point is a simulation conducted under present day orbital parameters.
 180 Lines represent linear regressions conducted for each pCO₂ level, the correlation coefficient of which
 181 is provided at the top of the figure using the same color code.

22. This raises the question of whether it was necessary to run the four simulations with different reconstructions. How different would the results have been if an aquaplanet simulation had been used and compared with the lithologic indicators in different positions according to the four paleogeographic scenarios?

The reviewer raises the question of the merits of testing four continental configurations versus a simpler modelling effort using only an aquaplanet configuration. We disagree with this approach, and do not think that using an aquaplanet would shed further light on the early Cambrian Earth System. We think that an aquaplanet configuration would lead to very different results, and that efforts are better put towards testing plausible continental configurations. This is because land masses impact both global and regional climate.

At the global scale, land masses modify the Earth's surface albedo, and consequently the radiative balance of the planet and the global surface temperature. Land masses also

impact surface- and deep-ocean circulation, which in turn affects oceanic heat transport and the latitudinal temperature gradient, again contributing to the global climatic state. See for instance Pohl et al. (2014) for a study quantifying the impacts of the position of continental blocks on global climate during the Ordovician – a period of time characterized by a similar continental configuration to that of the standard Cambrian configurations.

At the regional scale, the land-sea mask strongly impacts climate. Land has a much smaller heat capacity than ocean, so temperature swings over land are much larger and more in phase with the incoming solar radiation. This “continentality” effect leads to “continental climates”, as opposed to “oceanic climates”, and explains the typical temperature gradient observed moving inland from coastlines: cooler (warmer) summer (winter) temperatures in areas closer to the ocean. The differential behaviour of land and ocean, and the resulting seasonal temperature gradients, are also responsible for larger scale climatic features such as the monsoon. It should also be reminded that topography on land impacts precipitation patterns, triggering very localized “orographic” precipitations.

For these reasons, the climate simulated using an aquaplanet would be very different in both global and local aspects to the climates simulated using the four plausible continental configurations we test. The aim of this study is to see which combination of boundary conditions, including continental configuration, best agrees with the climatically sensitive geological data, as measured by our scoring system. The systematically different data/model agreement scores of the different configurations underlines the impact of continents on simulated climates at global and regional scales. Adding an aquaplanet experiment, whilst potentially interesting, would not shed any further light on Earth System boundary conditions in the early Cambrian, and would not impact the conclusions of this study, or help to clarify its message.

23. Another way of answering this is to make a plot like Fig. 3 showing the distribution of Köppen-Geiger climate regimes using model output (instead of lithologic indicators). This would make a nice addition to the paper.

We thank the reviewer for this interesting suggestion. In the revised version of the manuscript, we have included these additional figures in the supplementary file – one figure for each of the simulation conditions, similar to the Köppen-Geiger maps – as this may be of interest to future workers. However, we would again like to emphasize that this study revolves around regional, not latitudinal, climate patterns.

References cited in the response:

- Bertrand-Sarfati, J., Moussine-Pouchkine, A., Amard, B. and Ahmed, A.A.K. 1995. First Ediacaran fauna found in western Africa and evidence for an Early Cambrian glaciation. *Geology*, **23**, 133–136, [https://doi.org/10.1130/0091-7613\(1995\)023<0133:FEFFIW>2.3.CO;2](https://doi.org/10.1130/0091-7613(1995)023<0133:FEFFIW>2.3.CO;2).
- Chaboureau, A.-C., Sepulchre, P., Donnadieu, Y. and Franc, A. 2014. Tectonic-driven climate change and the diversification of angiosperms. *Proceedings of the National Academy of Sciences*, **111**, 14066–14070, <https://doi.org/10.1073/pnas.1324002111>.
- Chumakov, N.M. 2007. Climates and climate zonality of the Vendian: geological evidence. *Geological Society, London, Special Publications*, **286**, 15–26, <https://doi.org/10.1144/SP286.2>.
- Donnadieu, Y., Puc at, E., Moiroud, M., Guillocheau, F. and Deconinck, J.F. 2016. A better-ventilated ocean triggered by Late Cretaceous changes in continental configuration. *Nature Communications*, **7**, <https://doi.org/10.1038/ncomms10316>.
- Dufresne, J.-L., Foujols, M.-A., et al. 2013. Climate change projections using the IPSL-CM5 Earth System Model: from CMIP3 to CMIP5. *Climate Dynamics*, **40**, 2123–2165, <https://doi.org/10.1007/s00382-012-1636-1>.
- Foster, G.L., Royer, D.L. and Lunt, D.J. 2017. Future climate forcing potentially without precedent in the last 420 million years. *Nature Communications*, **8**, 14845, <https://doi.org/10.1038/ncomms14845>.
- Gibbs, M.T., Rees, P.M., Kutzbach, J.E., Ziegler, A.M., Behling, P.J. and Rowley, D.B. 2002. Simulations of Permian Climate and Comparisons with Climate-Sensitive Sediments. *The Journal of Geology*, **110**, 33–55, <https://doi.org/10.1086/324204>.
- Hutchinson, D.K., Coxall, H.K., et al. 2020. The Eocene-Oligocene transition : a review of marine and terrestrial proxy data , models and model-data comparisons. *Climate of the Past Discussions*, 1–71.
- Ladant, J.B., Donnadieu, Y. and Dumas, C. 2014. Links between CO₂, glaciation and water flow: reconciling the Cenozoic history of the Antarctic Circumpolar Current. *Climate of the Past*, **10**, 1957–1966.
- Landing, E., Geyer, G. and Brasier, M.D. 2013a. Cambrian Evolutionary Radiation: context, correlation, and chronostratigraphy—overcoming deficiencies of the first appearance datum (FAD) concept. *Earth-Science Reviews*, **123**, 133–172, <https://doi.org/10.1016/j.earscirev.2013.03.008>.
- Landing, E., Westrop, S.R. and Bowring, S.A. 2013b. Reconstructing the Avalonia palaeocontinent in the Cambrian: A 519 Ma caliche in South Wales and transcontinental middle Terreneuvian sandstones. *Geological Magazine*, **150**, 1022–1046, <https://doi.org/10.1017/S0016756813000228>.

- Laugié, M., Donnadieu, Y., Ladant, J.B., Mattias Green, J.A., Bopp, L. and Raisson, F. 2020. Stripping back the modern to reveal the Cenomanian-Turonian climate and temperature gradient underneath. *Climate of the Past*, **16**, 953–971, <https://doi.org/10.5194/cp-16-953-2020>.
- Licht, A., van Cappelle, M., et al. 2014. Asian monsoons in a late Eocene greenhouse world. *Nature*, **513**, 501–506.
- Pohl, A., Donnadieu, Y., Le Hir, G., Buoncristiani, J.F. and Vennin, E. 2014. Effect of the Ordovician paleogeography on the (in)stability of the climate. *Climate of the Past*, **10**, 2053–2066.
- Pohl, A., Donnadieu, Y., et al. 2020. Carbonate platform production during the Cretaceous. *GSA Bulletin*, 1–21, <https://doi.org/10.1130/B35680.1/4980955/b35680.pdf>.
- Scotese, C.R. and Wright, N. 2018. PALEOMAP Paleodigital Elevation Models (PaleoDEMS) for the Phanerozoic (PALEOMAP Project, 2018)<https://www.earthbyte.org/paleodem-resource-scotese-and-wright-2018/>.
- Sepulchre, P., Caubel, A., et al. 2020. IPSL-CM5A2 – an Earth system model designed for multi-millennial climate simulations. *Geoscientific Model Development*, **13**, 3011–3053, <https://doi.org/10.5194/gmd-13-3011-2020>.
- Zelinka, M.D., Myers, T.A., et al. 2020. Causes of Higher Climate Sensitivity in CMIP6 Models. *Geophysical Research Letters*, **47**, 1–12, <https://doi.org/10.1029/2019GL085782>.
- Zhu, J., Poulsen, C.J. and Otto-bliesner, B.L. 2020a. High climate sensitivity in CMIP6 model not supported by paleoclimate. *Nature Climate Change*, **1**, 1–2, <https://doi.org/10.1038/s41558-020-0764-6>.
- Zhu, J., Poulsen, C.J., Otto-Bliesner, B.L., Liu, Z., Brady, E.C. and Noone, D.C. 2020b. Simulation of early Eocene water isotopes using an Earth system model and its implication for past climate reconstruction. *Earth and Planetary Science Letters*, **537**, 116164, <https://doi.org/10.1016/j.epsl.2020.116164>.

REVIEWER COMMENTS

Reviewer #1 (Remarks to the Author):

The authors have done an excellent job in revising the paper. It is now much improved, and I recommend publication as is.

Reviewer #2 (Remarks to the Author):

Thanks to the authors for their detailed responses to my original comments on the manuscript. I appreciate that the authors are trying to demonstrate that a reconstruction based on paleomagnetic and paleobiogeographic data—rather than one based on climate sensitive lithologies—can reconcile the existing proxy data. From this standpoint, the study is interesting in that it shows that climate sensitive lithologies in and of themselves are not adequate constraints on paleogeography.

It would be helpful if the authors would briefly discuss in the manuscript body the differences in block positions between A, C, and D that lead to the disparate agreement scores and, specifically, the improved scores in D. In addition, the agreement scores for A, C, and D should be re-calculated after omitting the North China block to confirm that the absence of the block in B isn't leading to the high agreement.

My concerns about FOAM remain. The author's response to my comments about the FOAM ECS is confused and confirms my suspicions. The authors report a FOAM ECS of 2.55C at 8xPAL. This is a very low value at such high CO₂ levels. Note that the CMIP5 multimodel average is 3.25C at 1xPAL! Zhu et al. (2019) argued that the CMIP5 ECS is too low and showed that the CESM1.2 ECS is 4.2C at 1xPAL and rises to 6.6 and 9.7 C at 3xPAL and 6xPAL. In comparison, the FOAM ECS is 40% less at 1xPAL and has lower ECS values even at 128xPAL! The implication is that FOAM is not sensitive enough to CO₂ and can't be trusted to constrain ancient CO₂ levels. (CO₂ constraints based on FOAM should be considered maximum levels.)

Incidentally, Zhu et al. (2020) report that CESM2 has an ECS of 5.3 C at 1xPAL. The authors argue that this ECS is too high. Those findings are not relevant to this study, because FOAM has an ECS that is much, much lower at CO₂ levels up to 128xPAL.

The authors need to address the shortcomings of FOAM in the discussion. Lines 416-420 is not a very genuine attempt at this and is in fact duplicitous. For one, it's not okay to compare ECSs at different CO₂ levels because ECS changes with CO₂. The fact is that the FOAM ECS is very low, much lower than the CMIP ensemble average and that if the FOAM ECS was higher the estimates of the CO₂ range would change substantially.

REVIEWER COMMENTS

Our responses are in black type.

Reviewer #1 (Remarks to the Author):

1. The authors have done an excellent job in revising the paper. It is now much improved, and I recommend publication as is.

We are grateful to Reviewer #1 for their considered and constructive review which helped us to improve the clarity and focus of the manuscript. We are pleased to have satisfied their requirements for recommending publication.

Reviewer #2 (Remarks to the Author):

2. Thanks to the authors for their detailed responses to my original comments on the manuscript. I appreciate that the authors are trying to demonstrate that a reconstruction based on paleomagnetic and paleobiogeographic data—rather than one based on climate sensitive lithologies—can reconcile the existing proxy data. From this standpoint, the study is interesting in that it shows that climate sensitive lithologies in and of themselves are not adequate constraints on paleogeography. It would be helpful if the authors would briefly discuss in the manuscript body the differences in block positions between A, C, and D that lead to the disparate agreement scores and, specifically, the improved scores in D. In addition, the agreement scores for A, C, and D should be re-calculated after omitting the North China block to confirm that the absence of the block in B isn't leading to the high agreement.

We are grateful to Reviewer #2 for their recognition of the clarifications we have made to the manuscript. Inspired by the absence of the North China continent in configuration B, we conducted sensitivity analyses to assess the impact of removing whole continental blocks before scoring data-model agreement. Whilst the numerical agreement scores change, the overall patterns are robust to this type of systematic bias. We have added Figure S48 to the Supplementary Information (see below) and the following to the Methods (lines 496-501): “Without explanation, the North China palaeoplate was not included in configuration B (Landing *et al.* 2013a, b), despite having an extensive lower Cambrian stratigraphic record (e.g. Chough *et al.* 2010). We performed additional sensitivity analyses to examine the biasing effect of removing all North China geological data from the data-model agreement calculations. These sensitivity analyses show that, whilst systematic loss of geological data from one palaeocontinent changes the numerical scores, the overall patterns remain robust to this bias (Figure S48).”

Figure S48. Data-model agreement for all present day orbital configuration simulations scored against geological data excluding the North China palaeocontinent accounting for palaeogeographic uncertainty with a 250 km radius. The North China palaeocontinent is, inexplicably, not included in the original reconstructions of configuration B (Landing *et al.* 2013a, b), despite having extensive lower Cambrian deposits (e.g. Chough *et al.* 2010). Sensitivity analyses show that our results are robust to the systematic removal of all geological data from the North China palaeocontinent; whilst the numerical scores change, the overall patterns remain.”

3. *My concerns about FOAM remain.*

We appreciate Reviewer #2's concerns about FOAM, but strongly disagree with their conclusions. FOAM is not the most up-to-date climate model available, *but it is the most appropriate to the task at hand*. We are assessing a wide range of first-order constraints on boundary conditions including the positions of major continental blocks which in some cases vary from equatorial to polar positions. A higher throughput model is appropriate for simulating climatic conditions for the plausible range of boundary conditions hypothesized for the Cambrian world. A state-of-the-art, more computationally intensive, CMIP model would increase the precision of the climate picture from each individual simulation. But crucially it would not allow us to explore the range of plausible boundary conditions as effectively, and therefore it would confer little in the way of increased accuracy of our understanding of the Cambrian world.

4. *The author's response to my comments about the FOAM ECS is confused and confirms my suspicions. The authors report a FOAM ECS of 2.55C at 8xPAL. This is a very low value at such high CO2 levels. Note that the CMIP5 multimodel average is 3.25C at 1xPAL! Zhu et al. (2019) argued that the CMIP5 ECS is too low and showed that the CESM1.2 ECS is 4.2C at 1xPAL and rises to 6.6 and 9.7 C at 3xPAL and 6xPAL. In comparison, the FOAM ECS is 40% less at 1xPAL and has lower ECS values even at 128xPAL! The implication is that FOAM is not sensitive enough to CO2 and can't be trusted to constrain ancient CO2 levels. (CO2 constraints based on FOAM should be considered maximum levels.) Incidentally, Zhu et al. (2020) report that CESM2 has an ECS of 5.3 C at 1xPAL. The authors argue that this ECS is too high. Those findings are not relevant to this study, because FOAM has an ECS that is much, much lower at CO2 levels up to 128xPAL.*

The pre-industrial/Cambrian comparison made by Reviewer #2 is not the like-for-like comparison it is presented as. Solar luminosity was substantially reduced in the Cambrian compared to recent times, so 1 PAL $p\text{CO}_2$ simulations using Cambrian and pre-industrial boundary conditions do not result in similar climatic conditions across these different intervals. First-order energy balance requirements mean that a CO_2 -equivalent greenhouse effect of at least 8 PAL is required to offset the effect of decreased solar luminosity in the Cambrian to achieve a pre-industrial average global temperature. It would be surprising if 1 PAL $p\text{CO}_2$ forcing in Cambrian and pre-industrial times had similar ECS values.

We do not know where the values supporting the statement that "*the FOAM ECS is 40% less at 1xPAL*" derives from. FOAM simulations conducted for the purpose of this response, using a preindustrial palaeogeographic configuration and $p\text{CO}_2$ values of 1 PAL to 4 PAL (standard CMIP ECS assessment conditions), provide an ECS value of 4.04 K. This value is comparable to the mean of CMIP6 models (Zelinka *et al.* 2020), although significantly lower than the CMIP6 upper end-member ECS value of 5.3 K provided by CESM2 (a value that is too high to fit Early Eocene Climate Optimum proxy data (Zhu *et al.* 2020)).

Finally, as already stated in our first response to Reviewer #2, FOAM has been used with good effect to simulate more recent (Mesozoic and Cenozoic) greenhouse climates in a number of studies (e.g. Chaboureaud *et al.* 2014; Ladant *et al.* 2014; Licht *et al.* 2014; Donnadieu *et al.* 2016; Pohl *et al.* 2020) and performs well in an Eocene comparison project

with more advanced higher resolution GCMs (Hutchinson *et al.* 2021). FOAM is not a state-of-the-art climate model, but it most definitely does not perform as poorly as Reviewer #2 suggests.

5. *The authors need to address the shortcomings of FOAM in the discussion. Lines 416-420 is not a very genuine attempt at this and is in fact duplicitous. For one, it's not okay to compare ECSs at different CO2 levels because ECS changes with CO2. The fact is that the FOAM ECS is very low, much lower than the CMIP ensemble average and that if the FOAM ECS was higher the estimates of the CO2 range would change substantially.*

We are happy to expand on the limitations of FOAM in our Discussion and throughout the text, but we stand by our statement that FOAM is the right choice of climate model for the current task. As we have shown above (point 4), FOAM ECS is not “*substantially lower than the CMIP ensemble average*” when tested under comparable pre-industrial conditions. However, Cambrian and pre-industrial conditions are not closely comparable to each other in $p\text{CO}_2$ levels, as noted by the reviewer, or in solar luminosity, and these differences hinder comparison of ECS values in our study and those used in CMIP5 and CMIP6. We agree that different model ECSs will result in support for a different range of $p\text{CO}_2$ values and are happy to expand on this throughout the manuscript.

Introduction. Edited lines 104-112:

“The FOAM GCM is widely applied in deep-time palaeoclimate studies (Pierrehumbert 2004; Nardin *et al.* 2011; Pohl *et al.* 2016; Saupe *et al.* 2020), and is particularly well-suited to our purpose because its high throughput means that it can be used to produce the large numbers of simulations needed to test the wide range of boundary conditions plausible for the early Cambrian. However, as with most earlier generation GCMs, FOAM has a relatively low equilibrium climate sensitivity and consequently tends to simulate colder average temperatures for the same $p\text{CO}_2$ forcing than most recent models used in the Coupled Model Intercomparison Project (CMIP) (Hutchinson *et al.* 2021). Therefore, we hope that future work will be possible using CMIP-class models in the early Palaeozoic context, when Earth System boundary conditions for this interval are better constrained.”

Discussion. Edited lines 325-330:

“Because simulated temperatures are a function of the model climate sensitivity, we remind the reader that the numerical $p\text{CO}_2$ estimates are model-dependent. Because FOAM simulations are commonly slightly colder than those of more recent models for the same $p\text{CO}_2$ level (e.g. Hutchinson *et al.* 2021), using a GCM of the latest generation would likely simulate similar climatic states with lower $p\text{CO}_2$ forcing. Our $p\text{CO}_2$ values can therefore be considered an upper estimate. Nevertheless, the magnitude of early Cambrian atmospheric $p\text{CO}_2$ levels predicted by long-term carbon cycle models (Goddéris *et al.* 2014; Royer *et al.* 2014; Krause *et al.* 2018; Lenton *et al.* 2018) agrees with that suggested by our coupling of GCM and lithology data (16 to 32 PAL atmospheric $p\text{CO}_2$).”

Methods. Edited lines 428-430 (previously lines 416-420):

“**Note** that, as in any modelling study, the CO_2 -temperature relationship is dependent on the model climate sensitivity. **Using** a GCM with a **different** climate sensitivity would result in support for a different range of **numerical** $p\text{CO}_2$ values.”

References cited in the response:

- Chaboureaud, A.-C., Sepulchre, P., Donnadieu, Y. and Franc, A. 2014. Tectonic-driven climate change and the diversification of angiosperms. *Proceedings of the National Academy of Sciences*, **111**, 14066–14070, <https://doi.org/10.1073/pnas.1324002111>.
- Chough, S.K., Lee, H.S., et al. 2010. Cambrian stratigraphy of the North China Platform: revisiting principal sections in Shandong Province, China. *Geosciences Journal*, **14**, 235–268, <https://doi.org/10.1007/s12303-010-0029-x>.
- Donnadieu, Y., Puc at, E., Moiroud, M., Guillocheau, F. and Deconinck, J.-F. 2016. A better-ventilated ocean triggered by Late Cretaceous changes in continental configuration. *Nature Communications*, **7**, 10316, <https://doi.org/10.1038/ncomms10316>.
- Godd eris, Y., Donnadieu, Y., Le Hir, G., Lefebvre, V. and Nardin, E. 2014. The role of palaeogeography in the Phanerozoic history of atmospheric CO₂ and climate. *Earth-Science Reviews*, **128**, 122–138, <https://doi.org/10.1016/j.earscirev.2013.11.004>.
- Hutchinson, D.K., Coxall, H.K., et al. 2021. The Eocene–Oligocene transition: a review of marine and terrestrial proxy data, models and model–data comparisons. *Climate of the Past*, **17**, 269–315, <https://doi.org/10.5194/cp-17-269-2021>.
- Krause, A.J., Mills, B.J.W., Zhang, S., Planavsky, N.J., Lenton, T.M. and Poulton, S.W. 2018. Stepwise oxygenation of the Paleozoic atmosphere. *Nature Communications*, **9**, 4081, <https://doi.org/10.1038/s41467-018-06383-y>.
- Ladant, J.-B., Donnadieu, Y., Lefebvre, V. and Dumas, C. 2014. The respective role of atmospheric carbon dioxide and orbital parameters on ice sheet evolution at the Eocene–Oligocene transition. *Paleoceanography*, **29**, 810–823, <https://doi.org/10.1002/2013PA002593>.
- Landing, E., Geyer, G. and Brasier, M.D. 2013a. Cambrian Evolutionary Radiation: context, correlation, and chronostratigraphy—overcoming deficiencies of the first appearance datum (FAD) concept. *Earth-Science Reviews*, **123**, 133–172, <https://doi.org/10.1016/j.earscirev.2013.03.008>.
- Landing, E., Westrop, S.R. and Bowring, S.A. 2013b. Reconstructing the Avalonia palaeocontinent in the Cambrian: A 519 Ma caliche in South Wales and transcontinental middle Terreneuvian sandstones. *Geological Magazine*, **150**, 1022–1046, <https://doi.org/10.1017/S0016756813000228>.
- Lenton, T.M., Daines, S.J. and Mills, B.J.W. 2018. COPSE reloaded: An improved model of biogeochemical cycling over Phanerozoic time. *Earth-Science Reviews*, **178**, 1–28, <https://doi.org/10.1016/j.earscirev.2017.12.004>.
- Licht, A., van Cappelle, M., et al. 2014. Asian monsoons in a late Eocene greenhouse world. *Nature*, **513**, 501–506, <https://doi.org/10.1038/nature13704>.

- Nardin, E., Godd ris, Y., Donnadi u, Y., Hir, G.L., Blakey, R.C., Puc at, E. and Aretz, M. 2011. Modeling the early Paleozoic long-term climatic trend. *Geological Society of America Bulletin*, **123**, 1181–1192, <https://doi.org/10.1130/B30364.1>.
- Pierrehumbert, R.T. 2004. High levels of atmospheric carbon dioxide necessary for the termination of global glaciation. *Nature*, **429**, 646–649, <https://doi.org/10.1038/nature02640>.
- Pohl, A., Donnadi u, Y., Le Hir, G., Ladant, J.-B., Dumas, C., Alvarez-Solas, J. and Vandenbroucke, T.R.A. 2016. Glacial onset predated Late Ordovician climate cooling. *Paleoceanography*, **31**, 800–821, <https://doi.org/10.1002/2016PA002928>.
- Pohl, A., Donnadi u, Y., et al. 2020. Carbonate platform production during the Cretaceous. *GSA Bulletin*, **132**, 2606–2610, <https://doi.org/10.1130/B35680.1>.
- Royer, D.L., Donnadi u, Y., Park, J., Kowalczyk, J. and Godd ris, Y. 2014. Error analysis of CO₂ and O₂ estimates from the long-term geochemical model GEOCARBSULF. *American Journal of Science*, **314**, 1259–1283, <https://doi.org/10.2475/09.2014.01>.
- Saupe, E.E., Qiao, H., et al. 2020. Extinction intensity during Ordovician and Cenozoic glaciations explained by cooling and palaeogeography. *Nature Geoscience*, **13**, 65–70, <https://doi.org/10.1038/s41561-019-0504-6>.
- Zelinka, M.D., Myers, T.A., et al. 2020. Causes of Higher Climate Sensitivity in CMIP6 Models. *Geophysical Research Letters*, **47**, e2019GL085782, <https://doi.org/10.1029/2019GL085782>.
- Zhu, J., Poulsen, C.J. and Otto-Bliesner, B.L. 2020. High climate sensitivity in CMIP6 model not supported by paleoclimate. *Nature Climate Change*, **10**, 378–379, <https://doi.org/10.1038/s41558-020-0764-6>.

REVIEWERS' COMMENTS

Reviewer #2 (Remarks to the Author):

I thank the authors for their edits to the manuscript and their more careful characterization of FOAM and their results. Our disagreement about FOAM sensitivity aside, this is an interesting and well written paper. I ask for one minor revision:

Line 104. "The FOAM GCM is widely..." Please change to "The FOAM GCM is frequently..." The FOAM GCM is not widely used, even among the paleoclimate community.